# BACKPROPAGATION-FREE LEARNING THROUGH GRADIENT ALIGNED FEEDBACKS

## ABSTRACT

Deep neural networks heavily rely on the back-propagation algorithm for optimization. Nevertheless, the global sequential transmission of gradients in the backward pass inhibits its scalability. The Direct Feedback Alignment algorithm has been proposed as a promising approach for parallel learning of deep neural networks, relying on fixed random feedback weights to project the error on every layer in a parallel manner. However, it notoriously fails to train networks that are really deep and that include compulsory layers like convolutions and transformers. In this paper, we show that alternatives to back-propagation may greatly benefit from local and forward approximation of the gradient to better cope with the inherent and constrained structure of such layers. This directional approximation allows us to design a novel algorithm that updates the feedback weights called GrAPE (GRadient Aligned Projected Error). A first set of experiments are carried out on image classification tasks with feedforward and convolutional architectures. The results show important improvement in performance over other backpropagation-free algorithms, narrowing the gap with backpropagation. More importantly, the method scales to modern and deep architectures like AlexNet, VGG-16 and Transformer-based language models where the performance gains are even more notable.

## 1 INTRODUCTION

After many decades, the back-propagation algorithm (Rumelhart et al., 1986) is still a cornerstone in training deep neural networks. Some important issues with this algorithm arose with the increase in depth. For most of them, the proposed solutions have induced transformations in the architectures, without modifying the algorithm itself. For instance, residual connections (He et al., 2016) along with Batch and Layer Norm (Ioffe & Szegedy, 2015; Ba et al., 2016) changed the archictecture design to stabilize the training. However, with the exponential growth of architectures, the computational cost of BP raises questions about its efficiency and the opportunity of exploring other approaches.

The memory footprint and the execution time clearly represent limitations when training modern architectures, and the BP algorithm significantly contributes to the high energy consumption, when training large or huge models. Reducing this burden is therefore an important challenge, as is using specialized hardware with limited resources. This paves the way to exploring alternative algorithms that better take into account environmental consequences and practical hardware considerations.

The study of the learning phase for biological neural networks emphasizes the differences with the BP algorithm (Grossberg, 1987). First, the exact weight symmetry between the forward and backward passes does not align with biological observations. Furthermore, the global transmission of errors and the sequentiality of updates also make it highly implausible in reality (Lillicrap et al., 2020). Although this is no critical issue for machine learning (ML), this mismatch may inspire other kinds of learning algorithms by investigating new learning dynamics (Amato et al., 2019; Baydin et al., 2022; Hinton, 2022; Lillicrap et al., 2020; Mostafa et al., 2018; Meulemans et al., 2020).

The BP essentially consists of two phases, the forward pass that computes the output and the backward pass which propagates the error back through the network (also called *feedback*) to update the parameters. Many recent approaches focus only on changing the the feedback phase. The goal is to simplify the weight update process, namely how information about errors or targets is communicated to each layer. Notably, the Feedback Alignment (FA) algorithm (Lillicrap et al., 2016) showed that random feedback paths could be used to train deep networks, without the need for symmetric

weights. Nøkland (2016) even showed that this idea could be used to avoid sequentiality, leading to a parallelized training of neural networks with the Direct Feedback Alignment (DFA) algorithm. These algorithms, though simple, have shown promising results with simple feed-forward architectures, but fail with more structured layers and therefore cannot be applied to deep convolutional networks (Bartunov et al., 2018; Launay et al., 2019; Moskovitz et al., 2018; Crafton et al., 2019). Some refinements have recently tried to reduce the gap of performance with the BP (Akrout et al., 2019; Xiao et al., 2018; Lansdell et al., 2020; Guerguiev et al., 2020; Kunin et al., 2020).

A complementary line of work makes the opposite choice. An approximation of the gradients derives from a modified forward pass using directional derivatives (Baydin et al., 2022; Silver et al., 2021; Margossian, 2019; Fournier et al., 2023). These methods rely on the Jacobian-vector product during the forward pass, reducing the backward pass to a straightforward update of the parameters. However, the high variance of the estimation inhibits its scalability even when local losses are introduced as a remedy (Ren et al., 2022; Belouze, 2022; Fournier et al., 2023).

Furthermore, recent developments propose a noteworthy alternative based on *forward only* algorithms, for which a second and modified forward pass replaces the standard backward phase. Seminal works in this field are the Forward-Forward algorithm (FF) and the Present the Error to Perturb the Input To Modulate the Activity learning rule (PEPITA) proposed by Hinton (2022); Dellaferrera & Kreiman (2022). These algorithms, though relatively new and not yet widely studied have exhibited promising results and similarities with adaptive-feedback-alignment algorithms (Srinivasan et al., 2023).

In this paper, we draw inspiration from these promising approaches to design a new training algorithm for deep neural networks and our contributions are the following:

- The Zoutendijk theorem on line search convergence allows us to reconsider the weakness of previous feedback alignment methods and to design a method to align the random matrices for different kind of architectures.

- The alignment of the feedback matrices uses a forward gradient estimate, which is carefully used to avoid possible perturbation due to the high variance.

- We introduce a learning algorithm called GrAPE (GRadient Aligned Projected Error) that can handle and scale to modern architectures like deep convolutional networks and Transformer-based models.

- Empirical results show very promising results with models like VGG16 (Simonyan & Zisserman, 2014), trained on CIFAR100. To the best of our knowledge, this the first time.

- GrAPE reaches training losses that are on par with BP. A convergence towards poorer solutions could explain the lower test performance, leaving room for further improvement.

In the next section, we introduce the notations and discuss related work. Then the GrAPE algorithm is described in section 3, just before the empirical results presented in section 4

## 2 BACKGROUND AND RELATED WORKS

Let $f(x; \theta)$ be a neural network with $L$ layers, where $x \equiv h_0$ is the input, $\theta$ denotes its parameters. The output $\hat{y}$ of the network is computed sequentially as $\hat{y} = \sigma_L(a_L)$, with $a_l = W_l h_{l-1}$ and $h_l = \sigma_l(a_l)$, where $\sigma_{l,l\in[1,L]}$ is a non-linearity[1]. Given a loss function $\mathcal{L}$, $\nabla\mathcal{L}_l \equiv \partial\mathcal{L}/\partial a_l$ denotes the gradient of the loss with respect to the pre-activation of a specific layer $l$. The BP aims at computing this term incrementally, starting from the output layer. In particular $e \equiv \nabla\mathcal{L}_L$ defines the error of the network made on the input $x$, and simplifies to $e = \hat{y} - y$ for the Mean Squared Error (MSE) loss. The update of the output layer can be written $\delta W_L = -\eta e h_{L-1}^\top$, for a learning rate $\eta$. With $\odot$ denoting the Hadamard product, the update for the other layers are computed as follows:

$$\delta W_l = -\eta \delta a_l^{\mathrm{BP}} h_{l-1}^T, \text{ with } \delta a_l^{\mathrm{BP}} = \nabla\mathcal{L}_l = \left(W_{l+1}^T \delta a_{l+1}\right) \odot \sigma_l'(a_l) \ \forall l. \tag{1}$$

Hence, the update of a layer $l$ depends on the error computation (or propagation) of all the $\{L, L - 1, ..., l+1\}$ subsequent layers. This involves the corresponding weight matrices to "transport" the error signal from a layer to the previous one, as illustrated in Figure 1. This symmetric (the same weight

---

[1]With this notation, we encompass both linear layers and convolutional layers, followed by a non-linearity.

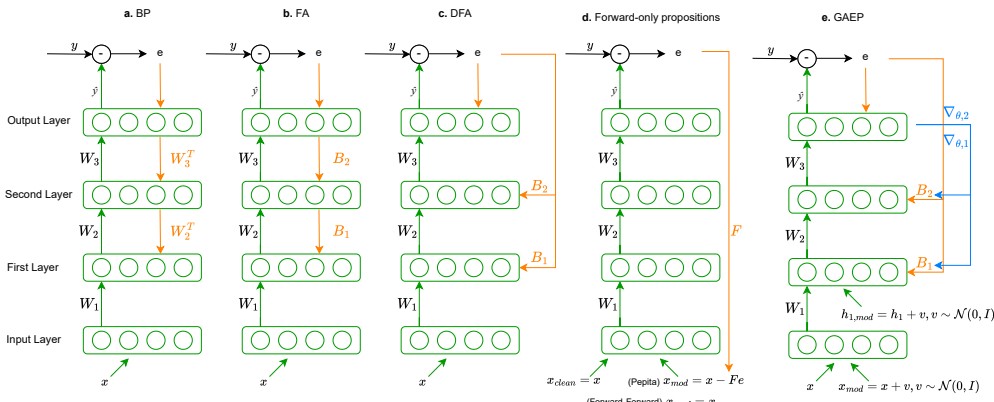

Figure 1: Overview of different error transportation configurations, inspired by Dellaferrera & Kreiman (2022). a) Back-propagation (BP). b) Feedback-alignment (FA). c) Direct feedback-alignment (DFA). d) Forward Only propositions (PEPITA and Forward-Forward). e) Gradient Aligned Projected Error (GrAPE). Green arrows indicate forward paths and orange arrows indicate error paths. Weights that are adapted during learning are denoted as $W_l$, and feedback weights are denoted as $B_l$ if specific to a layer (FA, DFA, GrAPE). The blue arrows indicate the activity perturbed forward gradient estimation $\mathbf{g_{a,l}(x)}$ as described by Ren et al. (2022).

matrix) and sequential nature of BP explains its limited effectiveness for parallelization. As state-of-the-art networks attain increasing size and depth, alternative methods that allow non-symmetric error transmission and enable parallelized training have emerged.

## 2.1 FEEDBACK ALIGNMENT (FA)

Feedback Alignment (FA) introduces a paradigm-shifting and biologically plausible alternative to gradient back-propagation (Lillicrap et al., 2016). At the core of this method, random feedback matrices drastically simplify the weight update process and break the symmetry. FA multiplies the output error by the random feedback matrices in order to obtain the error at each layer, which is in turn used to update the weights. The random matrices replace the term $\nabla \mathcal{L}_l$. However, the sequential aspect of the update remains, as the error is still propagated from layer to layer through the random feedback projections. With $B_l$ the fixed random feedback matrix of the $l$-th layer, the update can be estimated as follows:

$$\delta a_L = (B_L e) \odot \sigma'_L(a_L), \quad \forall l \in [1, L-1], \delta a_l = (B_l \delta a_{l+1}) \odot \sigma'_l(a_l)$$

This method draws its inspiration from biological neural networks, which do not exhibit symmetric weight transport during learning (Lillicrap et al., 2020), making this learning paradigm more biologically plausible. Nøkland (2016) went even a step further when presenting Direct Feedback Alignment (DFA). During training, the error signal is directly projected from the output layer to every hidden layers without modification or intermediate calculations. With this additional simplification, the updates can be easily parallelized.

$$\forall l \in [1, L], \delta a_l = (B_l e) \odot \sigma'_l(a_l) \tag{2}$$

FA and DFA have been shown to perform reasonably well on certain tasks and architectures, especially when considering its profound shift from the backpropagation method. While they do not consistently outperform or even compete with backpropagation, their simplicity along with biological plausibility stimulate research to scale up their use, as well as exploration to understand their key limitations. Bartunov et al. (2018) for example, show empirically that FA variants perform significantly worse on CIFAR-10 and ImageNet than BP, especially for convolutional networks. This is further analyzed by Launay et al. (2019) in which they exhibit a bottleneck effect that prevents learning in narrow layers, especially in the case of convolutional networks. As a workaround, some variants of FA showed promising performances on deep CNNs (Moskovitz et al., 2018). A seminal work by Akrout et al. (2019) for example used weight mirroring to adapt the feedback matrices during training, matching

BP performances. However these approaches stay sequential and similar approaches to DFA with target projection, for example DRTP (Frenkel et al., 2021) do not compete with BP on more complex convolutional networks.

In this paper, our ambition is to bridge the gap between the backpropagation and DFA for modern architecture, including convolution and tranformer layers. The work of Launay et al. (2020) showed promising results for transformer models trained with DFA. However, this is done according two settings: the 'macro' setting in which the feedback is applied after every encoder block and the 'micro' setting, in which it is done after every layer in those blocks. As explained by Launay et al. (2020) in Appendix D, backpropagation is still and always used through the attention mechanism itself, even in the 'micro' setting. Their training process therefore still relies on backpropagation within transformer layers, without reaching the same perplexity levels as full BP training. Our method allows us to better this proposition towards backpropagation-free training of modern architectures by providing blocks and/or layers a more useful feedback signal.

In their papers Nøkland (2016) and Refinetti et al. (2021) analyse the underlying dynamics in the FA-like algorithms to better explain their ability and unability to learn. A key lesson is that the angle between the feedback matrices and the true gradient must be lower than $\pm\pi/2$. Equivalently, if we denote $\omega_l$ this angle and $B_l$, the $l$-th layer's feedback matrix, Equation 3 must be fulfilled:

$$\forall l \in [1, L], \quad \cos(\omega_l) = \frac{\nabla\mathcal{L}_l^T B_l}{\|\nabla\mathcal{L}_l\| \cdot \|B_l\|} > 0 \tag{3}$$

We recognize a particular case of the Zoutendijk theorem (Nocedal & Wright, 1999), which ensures global convergence when the search direction makes an angle with the steepest descent direction bounded away from $\pi/2$. This theorem requires that the step length satisfies either the Goldstein or strong Wolfe conditions, and this is typically the case with standard learning rates. However, let us stress that the considered convergence is towards local minima and stationary points.

As previously mentioned, the recent work of (Akrout et al., 2019) revisits the idea to learn the feedback by emulating a Kolen-Pollack algorithm (Kolen & Pollack, 1994) or with an estimate of the transpose matrix. This idea facilitates the learning process of FA while reducing the angles $\omega_l$. This first attempt clearly shows that adaptive feedbacks enable the learning of networks on which FA previously failed. It also emphasizes the importance of Zoutendijk's theorem, even though the sequential learning process inherited from the FA still inhibits the potential improvements.

## 2.2 FORWARD ONLY CALCULATIONS

To design a backward free algorithm, a double forward method may provide a solution: the first forward pass is used to optimize the feedback matrices, while the second one compute the updates of the forward weights. The recent paper (Srinivasan et al., 2023) follows this trend and exhibits similarities between two forward-only frameworks, Forward-Forward and PEPITA (Hinton, 2022; Dellaferrera & Kreiman, 2022). Furthermore, it shows that those algorithms can be well-approximated by a feedback-alignment algorithm with adaptive feedback (AF) weights, modulated by the upstream network weights. As illustrated Figure1, PEPITA learning rule essentially is: $\delta W_l = (h_l - h_l^{err}) \odot (h_{l-1}^{err\,T})$, with $h_0^{err} = x - Fe$, where $F$ can be interpreted as a feedback matrix on the input.

Focusing on forward calculations, recent works explore unbiased estimations of the gradients, thanks to directional derivatives (Fournier et al., 2023; Baydin et al., 2022). The Forward Gradient (FG) algorithm, introduced by Baydin et al. (2022) and Silver et al. (2021) employs Forward-Mode Automatic Differentiation (AD) as proposed in Margossian (2019) to estimate gradients without relying on backpropagation, but rather on the Jacobian-vector product. These gradients are then used to update the weights in a similar fashion as in standard BP but using the forward gradients instead of the backward ones.

Let $F : \mathbb{R}^m \to \mathbb{R}^n$ be a differentiable function, $\mathbf{v} \in \mathbb{R}^m$ be a vector and $\nabla F$ be the Jacobian of $F$, which is a matrix of size $n \times m$. Forward-Mode AD calculates the directional gradient $\mathbf{d} = \nabla F \cdot \mathbf{v}$ of $F$ along $\mathbf{v}$, evaluated at point $\mathbf{x}$: $\nabla F.\mathbf{v} \equiv \lim_{\delta \mapsto 0} \frac{F(\mathbf{x}+\delta\mathbf{v}) - F(\mathbf{x})}{\delta}$.

Our goal is to approximate the Jacobian of the loss function, with respect to the weights. Baydin et al. (2022) showed that the forward gradient of the loss with respect to the weights at layer $l$, evaluated at point $\mathbf{x} \in \mathbb{R}^n$, $\mathbf{g_{W,l}} : \mathbb{R}^{n \times m} \to \mathbb{R}^{n \times m}$ as: $\mathbf{g_{W,l}}(\mathbf{x}) = (\sum_{i=1}^{L} \frac{\partial \mathcal{L}(x)}{\partial W_i} V_i) \cdot \mathbf{V_l}$ is an unbiased estimator of the gradient $\nabla \mathcal{L}_l$, where every $\mathbf{V_l} \in \mathbb{R}^{n_l \times n_{l-1}}$ are sampled from a multivariate Gaussian.

However, Ren et al. (2022) highlights the poor scalability of such methods with respect to the the number of parameters. To mitigate this issue, they introduced the Activity-Perturbed Forward Gradient algorithm inspired by Le Cun et al. (1988) and Widrow & Lehr (1990). A perturbation vector of the activations $\mathbf{U}_l \in \mathbb{R}^{n_l}$ is drawn from a multivariate standard normal distribution for each layer $l \leq L$ and the activity-perturbed forward gradient $\mathbf{g_{a,l}}(x)$ of the loss with respect to the $l$-th layer is estimated by Forward-Mode AD as:

$$\mathbf{g_{a,l}}(\mathbf{x}) \equiv \frac{\partial \mathbf{h}_l}{\partial \mathbf{W}_l} \sum_{i=1}^{L} (\frac{\partial \mathcal{L}(x)}{\partial h_i} \mathbf{U}_i) \otimes \mathbf{U}_l. \tag{4}$$

As the number of neurons $n$ is usually considerably lower than the number of weights, this method reduces the number of derivatives to estimate, thus reducing the variance in the gradient estimation as the guessing space goes from $O(n^2)$ to $O(n)$. However, the authors showed that the variance of the estimation is still high and increases with the number of neurons of the network. They propose the introduction of local losses to improve the quality of estimates. In Fournier et al. (2023), the authors notably extend this model with small local models and local auxiliary losses to provide a better estimation of the forward gradients than uncorrelated noise. In order to avoid relying on BP to learn the local models however, we will rely on the estimation given in equation 4 in our method.

It has to be noted that Forward-Mode AD uses dual numbers theory, meaning the FG calculations are theoretically done in parallel during the forward phase. The implementation proposed by Baydin et al. (2022) showed a computational overhead of 43% over standard forward phase, and a running time between 0.6 to 0.8 the time of full BP (forward+backward). Since then however, Pytorch has released a beta version of their implementation of Forward-Mode AD but no reliable information is available on the overhead of the forward phase. The implementation could also be optimized by using a dedicated library, such as is done by Ren et al. (2022) where they use JAX, which is known to be more efficient than Pytorch for this kind of computation.

## 3 GRADIENT ALIGNED PROJECTED ERROR (GRAPE)

In this section, we describe our main contribution: the GrAPE algorithm (Gradient Aligned Projection Error). Our contribution relies on the efficient use of forward gradients of each layer to adapt the feedback matrices. By combining both of these ingredients, the algorithm ensures the learning ability on more challenging tasks and with more complex models.

### 3.1 LIMITATIONS OF FIXED RANDOM FEEDBACK MATRICES

In the context of line search algorithms, the angle between the search direction and the negative gradient direction is a crucial factor. As mentionned in Section 2, the analogy for DFA is between the feedback path with and the gradient. This standard optimization problem is also pivotal in FA-like methods but the standard methods do not have access to the necessary information regarding the gradient. In the case of convolutional layers, the operation can be written as a block Toeplitz matrix (d'Ascoli et al., 2019). This constrained structure is impossible to reproduce with a fixed randomly sampled feedback matrix. This argument, developed by Refinetti et al. (2021), explains the failure of vanilla DFA on convolutional networks as shown by Launay et al. (2019), the convolutional weights being unable to capture the projected information correctly. However, if the projected information was aligned with the gradients of the corresponding weights, this issue could be resolved.

### 3.2 USE OF FORWARD GRADIENTS ESTIMATIONS

In this work, we use foward estimated gradient information to align the search direction and the gradient direction. More precisely, we extend the DFA framework to explicitly align the feedback matrices with the estimated gradient direction of each layer. The Zoutendijk theorem ensures

convergence as we constrain the search directions to be parallel to an unbiased estimate of the gradient.

More specifically, we use the activity perturbed forward gradient $\mathbf{g_{a,l}}$ of every layer defined by Equation 4 to update the corresponding feedback matrix towards the right direction before updating the forward weights using the DFA learning rule defined by Equation 2. Figure 1 represents our proposition along with other learning algorithm to better stress the key differences. By rewriting Equation 3 with the forward gradients, we can define the angle $\bar{\omega}_l$ as the angle between the forward gradient and the feedback matrix of the $l$-th layer, thus having:

$$\forall l \in [1, L], \cos(\bar{\omega}_l) \equiv \frac{g_{a,l}{}^T B_l}{\|g_{a,l}\| \cdot \|B_l\|}.$$

Assuming that the forward gradient method is accurate enough, we can ensure the alignment of the feedback matrices with the forward gradients, hence unlocking the learning. This assumption will be verified experimentally in Section 3.4. Therefore the learning condition of Equation 3 can be simply rewritten as following, discarding the sequential calculation of the true gradients:

$$\forall l \in [1, L], \quad \cos(\bar{\omega}_l) > 0, \tag{5}$$

### 3.3 LEARNING RULE AND ALGORITHM

Given the previous learning condition, we propose a new update rule for the direct feedback matrices to align their directions with the corresponding gradients. The goal of this update is thus to minimize the orthogonal directions while improving the gradient alignment:

$$\mathbf{B}_l^{t+1} \leftarrow \mathbf{B}_l^t - \eta_{B_l}(1 - \cos(\bar{\omega}_l)) * \mathbf{B}_l^t, \tag{6}$$

where $0 < \eta_{B_l} \leq 1$ plays the role of the learning rate associated with the feedback-matrix. This update ensures for each layer that parallel directions with the gradients are promoted for the error projection. Note that the feedback matrices could be scaled by taking into account the respective gradients' norms in order to respect the magnitude of the update that would have been conveyed by gradient descent. We leave this idea for future works, only ensuring in the present work a constant norm of each feedback matrices by a simple rescaling of the matrices at each iteration [2]. The complete learning method is summarized in Algorithm 1.

### 3.4 ALGORITHMIC DESIGN CHOICES

In this section, we discuss the validity of our algorithmic design choices and assumptions. The cornerstone concerns the gradients approximations using forward calculations. Baydin et al. (2022) and Ren et al. (2022) showed that an unbiased estimation of the gradients could be obtained using forward gradients calculated thanks to jacobian vector products. However, if we consider the original JVP formulation, the expectation of the forward gradient can be written as follows:

$$\mathbb{E}(g_{a,l}) = \mathbb{E}((\nabla\mathcal{L}_l U_l) \otimes U_l) = \mathbb{E}(U_l U_l^T)\nabla\mathcal{L}_l = Cov(U_l)\nabla\mathcal{L}_l.$$

For an unbiased estimate of the gradient, the covariance matrix of the perturbations must be the identity matrix in the subspace that the gradients lie in, for every layer. The choice of usable perturbations is thus rather limited. To meet this requirement, we draw perturbations from a multivariate standard normal distribution for each layer. This allows us to ensure an identity covariance matrix for the perturbations in Equation 4.

The high variance of forward gradients is the second challenge to address. This is especially the case for large networks as illustrated by Fournier et al. (2023). However, our method only relies on the direction of the gradients and not on their magnitude, meaning that if the cosine similarity between the forward gradient and the true gradient is high, our method should facilitate the learning. We experimentally verify this assumption in Figure 5, showing that the cosine similarity between the forward gradients and the true gradients is high enough to get useful information about the direction. One easy condition is to set the a large batch size. We will thus use a standard 256 batch size for our experiments when possible, downgrading to 128 for larger networks, due to computational constraints. We also mitigate this issue by averaging the gradients through the learning phase with a cumulative moving average, taking inspiration of the recycle procedure used by Miyato et al. (2018).

---

[2]This solution was already proposed in the original DFA learning rule (Nøkland, 2016)

---

**Algorithm 1** GrAPE algorithm

1: **Input :** Training data $\mathcal{D}$
2: Randomly initialize weights $w_{ij}^{(l)}$ for all $l$, $i$, and $j$.
3: Initialize $B_l$ for all $l$.
4: For a number of epochs, do :
5: **for all x** in $\mathcal{D}$ **do**
6:     Set $\mathbf{h}^{(0)} = \mathbf{x}$ and $\mathbf{d}^{(0)} = 0$.
7:     **for** $l = 1$ to $L$ **do**
8:         Propagate the input signal forward :
9:           Pre-activation signal : $\mathbf{a}^{(l)} = \mathbf{W}^{(l)}\mathbf{h}^{(l-1)}$   Activation signal : $\mathbf{h}^{(l)} = \sigma(\mathbf{a}^{(l)})$.
10:         Directional derivative : $\mathbf{d}^{(l)} = (\mathbf{W}^{(l)}\mathbf{d}^{(l-1)}) \odot \sigma'(\mathbf{a}^{(l)})$.
11:         **if** $l < L$ **then**
12:           Sample normal perturbations : $\mathbf{v}^{(l)} \sim \mathcal{N}(0, \mathbf{I})$.
13:           $\mathbf{d}^{(l)} = \mathbf{d}^{(l)} + \mathbf{v}^{(l)}$.
14:         **end if**
15:     **end for**
16:     Compute $\mathbf{e} = \frac{\partial \mathcal{L}(\mathbf{x})}{\partial \mathbf{h}^{(L)}}$.
17: **end for**
18: Update output layer weights:
19: $\mathbf{W}^{(L)} = \mathbf{W}^{(L)} - \lambda \mathbf{e} \otimes \mathbf{h}^{(L-1)}$.
20: **for** $l = 1$ to $L - 1$ **do**
21:     Update $B_l$ using equation 6
22:     Update the parameters with the learning rate $\lambda$: $\mathbf{W}^{(l)} = \mathbf{W}^{(l)} - \lambda(\mathbf{e}B^{(l)} \odot \sigma'(\mathbf{a}^{(l)})) \otimes \mathbf{h}^{(l-1)}$.
23: **end for**

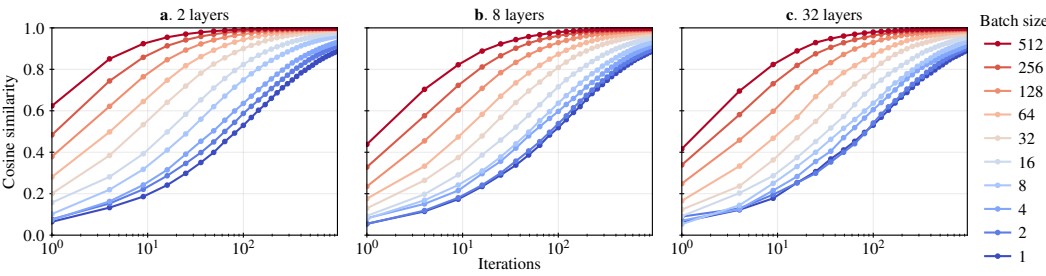

Figure 2: Cosine similarity between the forward gradients and the true gradients of the first layer for different batch sizes and network depths when averaging over the batch. Iterations indicate the number of JVP calculations executed for each batch. Synthetic data on feedforward neural networks was used for this experiment. From left to right, the rest of the networks are composed of 2, 8 and 32 layers. The batch size varies from 1 to 512.

## 4 EMPIRICAL EVALUATION

### 4.1 EXPERIMENTAL SETTING

A crucial aspect to BP alternatives is to ensure that they are rigorously implemented and evaluated on an equal footing and with the same conditions as BP. As they are still in their infancy when compared to BP, we believe it is critical for their development and future adoption to have a fair comparison. We thus implement our method on Biotorch (Sanfiz & Akrout, 2021) to make it accessible and to ensure the implementation does not contain any BP-related updates. We plan on releasing our code upon acceptance. It has to be noted that even though our algorithm allows a parallelized training of each layer, as modern deep learning frameworks are built for BP, the parallelization of the whole training process is not yet possible. This implementation heavily relies on `conv2d_input`, `conv2d_weight`, and `conv2d_bias` from the `torch.nn.grad` library to correctly adapt DFA and GrAPE to convolution related calculations.

We test our approach with simple 3 layers Feedforward Neural Networks (FFNNs), with hidden size 1024 and shallow convolutional neural networks on standard tasks, namely MNIST, CIFAR10 and CIFAR100. The shallow convolutional network follows the architectural details of LeNet-4 (LeCun et al., 1998). We compare our proposed GrAPE algorithm with standard FA and DFA, DRTP from Frenkel et al. (2021) and PEPITA from Dellaferrera & Kreiman (2022), with improvements recently developed by Srinivasan et al. (2023). However this promising algorithm is particularly young and does not yet apply to convolutional networks. For GrAPE, we draw the feedback weights from a normal distribution as they gave better results than using the uniform Kaiming initialization scheme and report the obtained results. Each network is trained for 100 epochs using Adam (Kingma & Ba, 2014) optimizer and the softmax cross-entropy loss function. Vanilla SGD is used to update the feedback weights. The initial learning rates used are $\lambda = 10^{-4}, \eta_{B_l} = 10^{-3}$ ; no further regularization or data augmentation was applied and a learning rate decay of 0.95 after every epoch was used. For PEPITA, we used the hyper-parameters provided by Srinivasan et al. (2023), noting that the performance get worse with more layers. All the experiments were done on a single GPU.

In Tables 2 and 3 we further extend GrAPE to deeper and more modern architectures, namely Alexnet, VGG-16 and Tranformer-based model. For this last instance we use the exact same experimental setup as proposed by Launay et al. (2020). We report in the Tables the average over 10 runs of the best test performance.

Table 1: Performances of a shallow convolutional network (CNN) and a 3 layer Multi Layer Perceptron (MLP) trained on the MNIST and CIFAR10 datasets with different learning algorithms (in percentages)

| Method | Parallelizable | MNIST | | CIFAR10 | | CIFAR100 | |
|---|---|---|---|---|---|---|---|
| | | MLP | CNN | MLP | CNN | MLP | CNN |
| BP | No | $98.73 \pm 0.04$ | $99.03 \pm 0.02$ | $54.09 \pm 0.14$ | $74.66 \pm 0.08$ | $28.18 \pm 0.45$ | $44.22 \pm 0.19$ |
| FA | No | $98.36 \pm 0.04$ | $98.7 \pm 0.07$ | $52.18 \pm 0.15$ | $71.05 \pm 0.18$ | $24.54 \pm 0.22$ | $35 \pm 0.27$ |
| DRTP | Yes | $95.7 \pm 0.12$ | $98.5 \pm 0.17$ | $47.55 \pm 0.12$ | $64.73 \pm 0.62$ | $18.63 \pm 0.43$ | $30.54 \pm 0.12$ |
| DFA | Yes | $98.21 \pm 0.07$ | $98.6 \pm 0.04$ | $51.32 \pm 0.32$ | $69.34 \pm 0.4$ | $22.44 \pm 0.23$ | $34.53 \pm 0.42$ |
| PEPITA | Yes | $98.01 \pm 0.08$ | NA | $52.01 \pm 0.13$ | NA | $21.87 \pm 0.25$ | NA |
| GrAPE (ours) | Yes | $98.35 \pm 0.02$ | $98.7 \pm 0.09$ | $\mathbf{52.8 \pm 0.02}$ | $\mathbf{72.07 \pm 0.13}$ | $\mathbf{25.02 \pm 0.63}$ | $\mathbf{37.21 \pm 0.43}$ |

Table 2: Performances of AlexNet and VGG-16 on CIFAR100.

| Method | AlexNet | VGG-16 |
|---|---|---|
| BP | $60.43\% \pm 0.35$ | $73.15\% \pm 0.61$ |
| DFA | $29.75\% \pm 0.58$ | $1\%$ |
| GrAPE (ours) | $\mathbf{40.43\% \pm 0.23}$ | $\mathbf{32.4\%}$ |

Table 3: Best validation perplexity after 20 epochs of a Transformer trained on WikiText-103 (lower is better).

| | DFA | GrAPe | BP |
|---|---|---|---|
| Macro | 52.0 | $\mathbf{45.6}$ | 29.8 |
| Micro | 93.3 | $\mathbf{84.8}$ | |

## 4.2 RESULTS ANALYSIS

The results reported in Tables 1, 2 and 3 show the superiority of our method over the other local and parallel learning approaches. First of all, GrAPE clearly betters the performances of DFA when it is possible on small networks, showing that the alignment of the feedback matrices with the forward gradients is a promising way of improving the learning of FA-like algorithms. It also helps bridging the performance gap of parallel learning algorithms with BP, beating DRTP, PEPITA and DFA on every tasks. However our results show that we are still lagging behind BP even on small networks.

Our method also shows promising results on larger networks, such as AlexNet, VGG-16 and even a Transformer. The performance gap with DFA is here even more notable. The training of deep convolutional networks like VGG-16, which was showed to be untrainable through DFA by Launay et al. (2019) can be achieved with our method, unlocking new architectures for parallel learning algorithms. On the other hand, the gap with BP is still important, showing room for major improvement. We note a positive effect of batch normalization and the impact of the feedback initialization shows us that, even though the effect of these operations are well-studied for standard BP, much work is yet needed to understand their way of functioning and optimize them for feedback learning.

When it comes to training a Transformer, we adopt the same paradigm as done by Launay et al. (2020). Following their training principles, we trained under two settings: *Macro* where the feedback is applied after every transformer layer, implying BP inside the layer, and *Micro* where the feedback is applied after every sub-layer. We reported the results on DFA with the best hyper-parameters found in the original paper and used the same setting to train with the GrAPE learning rule. The results presented in Table 3 show a clear improvement of GrAPE over DFA when training modern architectures, however still largely lagging behind BP in terms of validation performance.

We illustrate this in Figure 3. There is a wider gap between the training loss and the test loss for GrAPE than for the other learning algorithms. While GrAPE still largely surpasses DFA as seen in Table 2, the loss curves indicate that GrAPE performances on the training set are on par with BP, and that GrAPE suffers from more over-fitting than the other algorithms. A thorough analysis of how to transpose standard and widely studied regularization techniques from BP to GrAPE could thus vastly reduce the generalization gap of our method. In this way, the impact of standard hyper-parameters such as activation functions, is also to be studied, which we leave for future work.

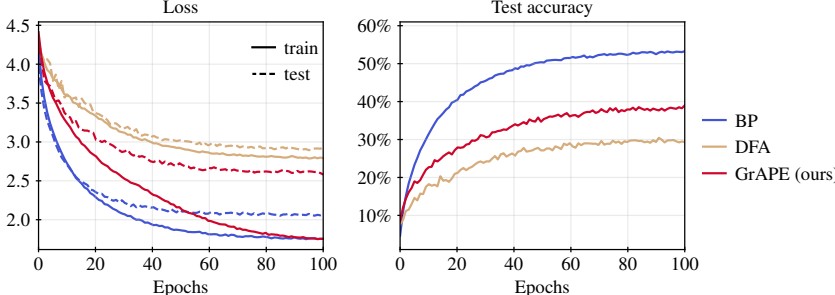

Figure 3: Training Dynamics of BP, DFA and GrAPE when training AlexNet on CIFAR 100 with batch size 128.

## 5 LIMITATIONS AND DISCUSSION

Experimental results show that the alignment of the feedback matrices with the forward gradients is a promising way of improving the learning of FA-like algorithms. This alignment allows the feedback to follow directions nearly parallel to the gradient direction. This makes the difference between GrAPE and previous work, when we consider deep architectures such as AlexNet and VGG16 on CIFAR100 (Bartunov et al., 2018; Launay et al., 2019).

Our approach also performs better than PEPITA, even in its recent and augmented version (Srinivasan et al., 2023). Borrowing ideas from the seminal work of Akrout et al. (2019), their method can be considered as a tailored feedback-alignment algorithm with adaptive feedback weights. The major difference with our work is that we have a feedback matrix for each layer, which enables a local injection of the error signal. We can therefore use GrAPE on much deeper architectures, since this prevents the error signal to become more and more distorted through the second forward pass.

However our method is still lagging behind BP, especially on larger networks. A couple of decades were necessary to reach such performances with BP. Our line of work is much more recent and requires further improvements to bridge this gap. For instance, the poor generalization we observed in Figure 3 calls for the design of adapted regularization techniques. Furthermore, the high variance of forward gradients increases with the number of neurons in the network (Ren et al., 2022). Hence, the forward gradients differs from the true gradients, as showed in Section 3.4, thus perturbing the learning direction. This could be mitigated either with iterative calculations of the forward gradients, or with local losses Fournier et al. (2023), but at the expense of a higher computational cost. On the optimization side, it is worth noticing that the Zoutendijk theorem only ensures convergence towards a stationary point, which could be a saddle point. Our method could benefit from second order information, gathered for instance in the forward pass, to improve the convergence guarantee.

Lastly, we showed with this method that DFA could be adapted to take into account inherent gradient structure for particular calculations. In order to fully train modern architectures as Transformers with

a parallel algorithm, it would thus be interesting to take into account the structure of some underlying operations such as residual connections (He et al., 2016) and self-attention. The former was motivated to address the vanishing gradient issue (due to BP), while the latter is dedicated to structured inputs. We leave the adaptation of feedback-projection to these operations for future work. The presented results on Transformers Table 3 also opens up new avenues for parallel training of networks, dividing it into smaller chunks that would receive gradient-aligned feedback information.

## 6 Conclusion

The GrAPE algorithm presented in this work emerges as a promising local alternative to conventional BP. It successfully mitigates challenges posed to the DFA by using approximations of the gradient calculated during the forward phase and enables parallel updates of all the parameters through the network. We showed its ability to learn complex tasks with deep architectures. The adaptive feedback allows the training algorithm to align its direction with the gradients. Hence with GrAPE, we can reach training losses that are on par with BP, while the generalization capacity must be improved. By turning the backward pass into a feedback projection, the information propagation in GrAPE follows the forward-only paradigm. This interstingly brings our method closer to biological models of the brain and the real-time constraints inherent to neuromorphic systems.

Our findings also show that in order to scale parallel learning algorithms to modern networks, a theoretical and empirical work is still needed to understand how all the deep-learning ingredients that interact with the BP recipe can be also used with DFA-like algorithms. These ingredients include initialization schemes, architectural operations like Batch or Layer Normalization, as well as residual connexions and dropout.

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

# A APPENDIX

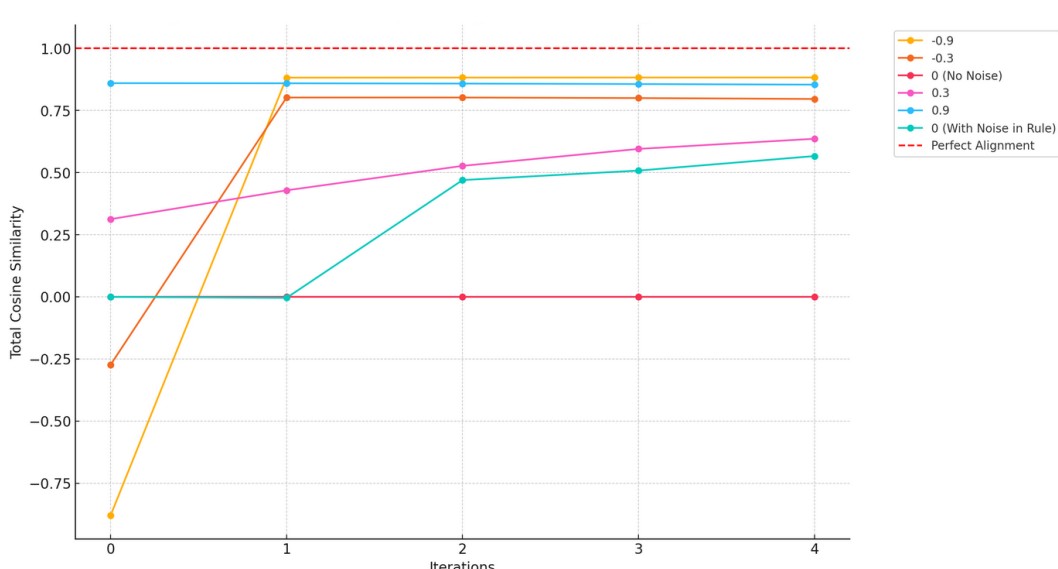

Figure 4: Cosine similarity variations between different matrix using learning rule Equation 6 and a fixed gradient matrix of sizes $(4 \times 4)$. In practice we add random noise of small magnitude with respect to the norm of the matrix to ensure updates even when cosine similarity is 0.

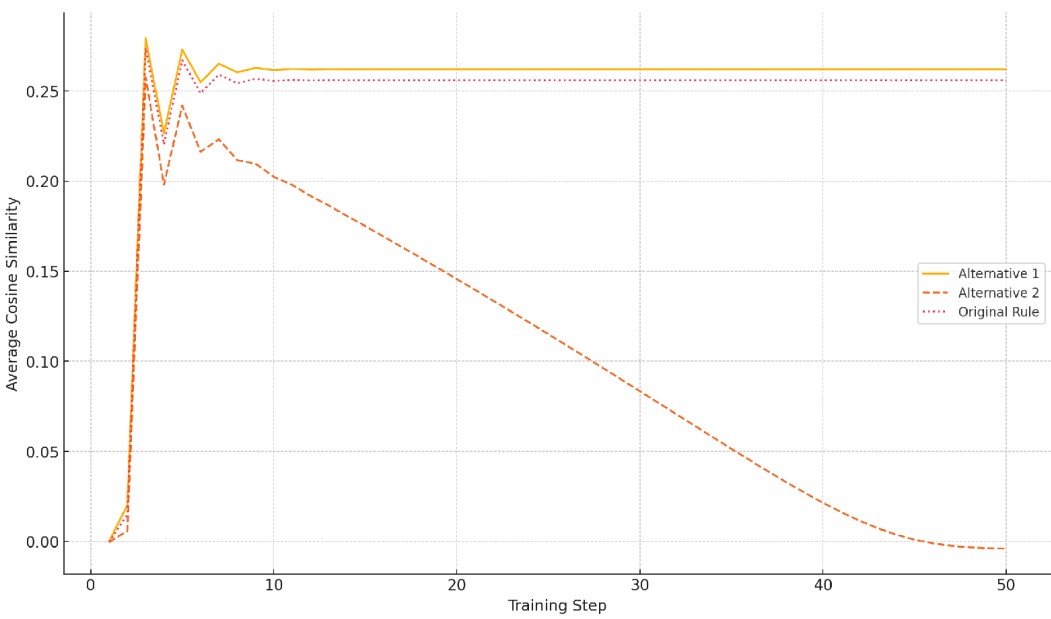

Figure 5: Comparison of the convergence of average cosine similarity of different learning rules over 50 training steps. Results are averaged over 10 runs.

