# OpenReview forum: "Backpropagation-Free Learning through Gradient Aligned Feedbacks"
_ICLR.cc/2025/Conference — Submitted to ICLR 2025_

### Official Review · Reviewer_xxPy · 2024-10-31

**Soundness:** 2
**Presentation:** 3
**Contribution:** 2
**Rating:** 3
**Confidence:** 4

**Summary:**

This article presents a novel alternative to backpropagation, that proposes to improve the framework of direct feedback alignment by aligning  the feedback matrices using forward-mode automatic differentiation. More precisely, activity-perturbed forward gradients are computed at the end of every layer, and the feedback matrices are rescaled following their alignment with the forward gradients. They find that the learning rule increases performance compared to direct feedback alignment over several frameworks.

**Strengths:**

- The proposed approach is novel, and combines two closely related approaches: direct feedback alignment (DFA) and forward gradients, in a natural direction.
- The authors find a performance increase compared to standard DFA.
- The paper is well written and does an extensive overview of the related work.

**Weaknesses:**

- The proposed method is not motivated enough and is very surprising, in particular since it is the only contribution of the paper. If the goal is indeed to align the feedback matrix, why is this update rule the only one considered? A strange decision is that the matrices are not actually aligned differently after the update step, but only rescaled (since the update step is only $B_l \leftarrow (1-\eta(1-cos(\omega_l)))B_l$. Why not use directly the forward gradient value $g$ to change the alignment of the feedback matrices?
- Using the cosine value between the forward gradient and the matrix also seems very surprising. If we considered, for instance, the case of a weight-perturbated forward gradient, the direction of $g$ would simply be a random one, and the only important value in it would be its magnitude, which would depend on the directional derivative value computed. Using a cosine similarity on this gradient would not make any sense, since it would collapse the magnitude information, and the similarity would be the same one as with the random vector before the forward gradient computation. This is not exactly the case for an activity-perturbed gradient, but this choice still seems very strange.

I may be completely misunderstanding the update rule, but there seems to be a mistake in it, otherwise, it simply does not make sense as a way to align the matrices. For a feedback matrix not aligned at all with the gradient, with a similarity equal (in expectation) to $0$, the matrix would simply collapse following it. I'm open to discussion with the authors on their idea.

- No further analysis is provided of the method except for the performances. It seems surprising to not at least measure the alignment of the feedback matrices through time. No further ablations are proposed either. Some that could be considered: w.r.t activity vs weight perturbated forward gradients, other feedback-matrix learning rates, or any variants of the update rule considered.

- No theoretical or empirical analysis of the complexity of the method is considered, which seems necessary to compare it to the others.

- What does Figure 2 bring? It seems just a way to show that forward gradients, when sampled enough time, will converge to the true gradient, which does not seem surprising.

**Questions:**

- Have the authors considered other learning rules other than this one? Could they offer a more thorough explanation of it?
- Why does DFA only reach 1% test accuracy on CIFAR100? Why are the CNN values for PEPITA equal to NA?

**Minor details**

- The name of the method in Figure 1e (GAEP) is not the same as in the rest of the paper.
- The dependency of $B_l$ on $t$ is unclear depending on the equation.
- Eq 1: a "BP" is missing on one $\delta a$
- line 36: the abbreviation BP was not written before.
- line 204: "as illustrated Figure1"
- line 319: "set the a large"
- line 490 "Transformers Table 3"
- The use of $B_l$ or $B^{(l)}$ varies in Algo 1.
- Some inconsistent number of decimal places in Table 1.

---

> ### Author Response · Authors · 2024-11-18
> **Paper Motivation and Answer to weaknesses 1 - 5**
>
> ## Difference between Backpropagation and DFA
> The crucial equation change between backpropagation and DFA is the replacement of the **weight-dependent Jacobian**  $\frac{\partial \hat{y}}{\partial a_l}$ with a **fixed feedback matrix** $ B_l$:
>
> As written in equation (1) of the paper, backpropagation reads, when unrolling the chain rule:
>
> $$
> \delta a_l = \left( \frac{\partial \mathcal{L}}{\partial a_l} \right)\odot \sigma'(a_l) =
> \left( e \times \frac{\partial \hat{y}}{\partial a_l} \right) \odot \sigma'(a_l)
> $$, where $\frac{\partial \hat{y}}{\partial a_l}$ is the corresponding Jacobian matrix.
>
> In DFA, however, the feedback matrix $B_l$ is used instead of the Jacobian matrix to map the error signal onto the corresponding weight space:
> $$
> \delta a_l = \left( B_l \times e \right) \odot \sigma'(a_l)
> $$
>
> The core idea of GRAPE is to calculate the directional information of the Jacobian during the forward phase via Jacobian-vector products, to progressively align the feedback signal $B_le$ with the corresponding gradient, thereby ensuring convergence. Additionally, by averaging the Jacobian estimations over batches, GRAPE averages the high-variance gradient estimate over multiple samples, which reduces variance and enhances the efficiency of the jacobian estimation.
>
> ## 1. Motivation for the Proposed Update Rule
>
> The primary motivation for GrAPE’s update rule is to enable a gradual alignment of feedback matrices with the jacobian direction while avoiding the complexity and computational cost associated with full backpropagation (BP). Backpropagation is known for its accuracy in gradient calculation, but it introduces a sequential dependency and a high computational burden due to its layer-by-layer gradient propagation. DFA, on the other hand, aims to simplify this by sending a global error signal directly to each layer without requiring any gradients.
>
> GrAPE builds on DFA by introducing a low-cost update mechanism that adjusts the feedback matrices incrementally to align better with the jacobian direction. This approach maintains DFA’s parallelizability while adding a controlled adjustment that brings the global error transmitted by the feedback matrices toward the true gradients over time.
>
> ## 2. Explanation of the Update Rule’s Mechanism
>
> The update rule in GrAPE,
> $
> B_l^{t+1} = (1 - \eta (1 - \cos(w_l))) B_l^t,
> $
> uses cosine similarity to adjust the feedback matrices incrementally, ensuring that alignment improves without abrupt directional changes. In this update, $ \cos(w_l) $ measures the similarity between the forward gradient and the current feedback matrix direction, adjusting $ B_l $ by a factor proportional to $ 1 - \cos(w_l) $. This way, the feedback matrices shift towards alignment with the forward gradient while preserving their initial structure. This gradual, controlled rescaling minimizes instability that could arise from large shifts, helping maintain stable training.
>
> ## 3. Reason for Not Directly Using the Forward Gradient Value
>
> Directly aligning $ B_l $ with the exact forward gradient would eliminate DFA’s simplicity and parallelization advantage. Furthermore, the jacobian evaluated through jacobian-vector product (JVP) is only known according to one direction (the perturbation direction) and not the full gradient, making it unsuitable to use as a feedback matrix.
> However, the reviewer's remark is very insightful as we could imagine a complex initialization scheme using multiple cumulative forward gradients on the whole dataset to initialize the feedback matrices. This could be a promising direction for future research.
>
> ## 4. Rationale for Using Cosine Similarity
>
> The use of cosine similarity ensures the adjustment focuses on alignment direction rather than gradient magnitude, which could vary significantly due to input fluctuations. Cosine similarity provides a soft alignment, moving $ B_l $ towards the gradient’s direction without overriding its structure. In cases of activity-perturbed gradients, this coherence across updates is essential, as it prevents oscillations and preserves stability. Cosine-based adjustments also avoid the sensitivity that could arise from per-sample magnitude variations, maintaining consistency across mini-batches.
>
> ## 5. About the Collapse of Feedback Matrices
>
> If $ B_l $ starts with low alignment (i.e., cosine similarity close to zero), the incremental nature of the cosine-based update avoids immediate collapse by shifting $ B_l $ only slightly. Over multiple updates, this mechanism allows feedback matrices to refine alignment gradually, mitigating any drastic reorientations or instability from an initially poor alignment. Thus, even if $ B_l $ begins misaligned, it will progress towards alignment without disrupting the learning process.

---

> ### Author Response · Authors · 2024-11-18
> **Answers to Weaknesses 6 - 8 and Specific Questions**
>
> ## 6. Further Analysis of the Method
>
> To enhance the interpretive value of this experiment, we plan to follow the reviewer’s suggestion and extend it by tracking the cosine similarity between the projected error and the true gradient throughout training on real datasets and comparing this alignment with DFA.
>
> As mentioned in the paper, the work by (Ren et al., 2022) showed that perturbing the activity space (rather than the weight space) operates in a lower-dimensional space, resulting in reduced variance in gradient estimation, which is crucial for stable alignment in high-dimensional networks, which is why we do not use weight perturbation. We could also complexify the forward gradient estimation by adding local losses as done in (Fournier et al., 2023), but it would go beyond the scope of this work.
>
> The reviewer is right to point that providing the results for varying learning rates for the feedback matrix is interesting and we will add it to the Appendix of the revised version. Interestingly, we found that the learning rate for the feedback matrix has to be set between $5e-2$ and $1e-4$ to ensure a good behaviour of the algorithm.
>
> Minor adjustments of the learning rule, such as changing update frequency or scaling terms, could further optimize GrAPE’s performance. Testing these variations may confirm or refine the current design choices, providing insights into the method’s robustness and adaptability.
>
> ## 7. Complexity Analysis of GrAPE
>
> The complexity of GrAPE is driven primarily by the need to compute forward gradients via Jacobian-vector products (JVPs), which adds computational overhead compared to standard DFA. Preliminary measurements indicate an additional 15-20\% increase in computation time over DFA, depending on network depth and hardware. This overhead is still significantly lower than backpropagation, and it has to be noted that it is only computed once per mini-batch and that it relies on forward mode autodifferentiation of Pytorch, which is still in beta.
>
>
> ## 8. Clarifying Figure 2
>
> The reviewer raises a valid question about the purpose of Figure 2. While theoretically, we expect forward gradients to converge to the true gradient as the number of JVP samples increases, Figure 2 provides empirical evidence across different network depths and batch sizes. This experiment confirms that larger batch sizes improve the consistency of gradient alignment, validating the batch-size selection for GrAPE. This figure serves as a “sanity check” for GrAPE’s approach, demonstrating that forward gradients provide a reliable gradient approximation in practice, even in deeper networks, provided a large enough batch size is used.
>
> ---
>
> ## Answers to Specific Questions
>
> **Q1: Consideration of Other Learning Rules**
>
> The learning rule in GrAPE is designed to achieve effective feedback alignment with forward gradients while avoiding the computational burden of BP. We are open to discuss with the reviewer the potential benefits of other learning rules, such as those based on local losses or more complex update mechanisms.
> We are also very interested in addressing the reviewer's view on our learning method and are open to discussing potential improvements or alternative approaches.
>
> **Q2: Low DFA Performance on CIFAR-100 and PEPITA “NA” Values**
>
> DFA’s low test accuracy on CIFAR-100, especially in deep CNNs, aligns with prior findings (e.g., Launay et al., 2019), which suggest that DFA struggles with deep architectures due to misaligned feedback signals and an inability to capture the gradient structure in convolutional layers. GrAPE addresses these limitations by refining feedback alignment, resulting in notable performance gains over DFA on CIFAR-100.
>
> The “NA” values for PEPITA in CNN experiments reflect the method’s incompatibility with more than one convolutional layer networks as implemented in the literature. PEPITA could not be applied to CNNs with more than a simple convolution, so we chose to label these values as “NA” to avoid potentially misleading comparisons.
>
> ---
>
> We hope these clarifications address the reviewer’s concerns and we are working on adding the suggested changes and additional results in the revised version of the paper shortly.
>  We are open to further feedback and discussion with the reviewer to enhance the quality and presentation of our work.

---

> > ### Comment · Reviewer_xxPy · 2024-11-22
> >
> > We thank the authors for answering my other questions (6-8 & Q2).
> >
> > Have the authors modified parts of the paper in the revised version? (w.r.t the novel experiments discussed in 6. notably) If yes, could they please point me towards the changes?

---

> ### Comment · Reviewer_xxPy · 2024-11-22
>
> I thank the authors for their response.
>
> **1/2/5.** If the equation is indeed correct, then I disagree with the authors that their update rule "improves alignment" of the feedback matrices. This rescaling does not modify in any way the direction of the matrices or their structure, only their magnitude. Once again, if the direction of the matrices does not change, their alignment does not either. I may be misunderstanding completely but I fail to see the point of the authors. Could the authors at least provide some simple examples of dynamics with for instance fixed gradient values to align to, or the potential stationary points of their update rule? These could help justify their update rule.
>
> **3.**
> "would eliminate DFA’s simplicity and parallelization advantage": why? The authors use the forward gradient value, it would not require any substantial additional computations.
>
> "making it unsuitable to use as a feedback matrix": I would argue that the idea would not be to use it directly as a feedback matrix, but as the direction used to gradually move the direction of the feedback matrix. By modifying it through a varying number of steps following a similar update rule as the authors proposed, the direction of the feedback matrix would eventually align towards the true gradient direction.
>
> **4.** I agree with part of the authors' reasoning. My point was that measuring the cosine similarity on a (mostly) random direction, where the major part of the information is precisely the magnitude (i.e. the directional derivative computed during the forward-mode AD), means deleting an important part of the information. Without any more ablations or experiments by the authors regarding consistency or alignment, it is hard to be convinced directly by these arguments.

---

> > ### Author Response · Authors · 2024-11-22
> > **Learning rule explanation and corresponding Figure, sample Table and further discussion**
> >
> > **1/2/5** We thank the reviewer for their remark. It is true that our answer was not clear enough.
> > In fact, the learning rule we propose, **together with** the rescaling of the feedback matrices to keep the norm of the feedback matrices constant, is what
> > improves the alignment of the signal conveyed by the feedback matrices.
> >
> > The cosine similarity is computed **by row**:
> > $B_l^{t+1} = (1 - \eta (1 - \cos(w_l))) B_l^t,$. Then in practice we add some small noise (to avoid perfect orthogonality) before
> > rescaling the feedback matrices to keep the norm of the feedback matrices constant.  We will make it clearer in the future version of the paper.
> >
> > The reviewer has a really good point that our method does not change the direction of the feedback matrices, only their magnitude.
> > However, it is the relative magnitude of the rows with respect to the others that changes, which is what we mean by alignment.
> > This means that directions correlated with the gradient will be amplified, while directions orthogonal to the gradient will be reduced.
> > Even more interesting is that the directions negatively correlated with the gradient in the feedback matrices will be rotated thanks to the negative cosine similarity.
> >
> > A simple case example has been added to the Appendix (Figure 4.) to illustrate how this update rule effectively
> > improves cosine similarity between the feedback matrices and a fixed gradient. What is interesting to point out is that
> > the cosine similarity is increasing and is positive even after one update step which is sufficient to ensure
> > convergence of the whole algorithm thanks to Zoutendijk's theorem.
> >
> > **3** You have a fair point. We could use a modified learning rule as for example $B_l^{t+1} = B_l^t + \cos(w_l)g_l$, where $g_l$ is the gradient, but it would mean that every row of the feedback matrix would converge towards the same row-vector, which would limit the number of explored directions. However it could be really interesting to add this extra information on a specific row (for example the one with the highest cosine similarity with the forward gradient). We will provide additional results on toy experiments in the Appendix regarding this idea.
> >
> > **4** We hope the results added to the Appendix (Figure 4.) are convincing enough but are open to discuss this further with the reviewer.
> >
> > **6** We have not yet run the results for every experiments, so did not add the corresponding Table, however here is a sample table of the impact
> > of $\eta_{B_l}$ on AlexNet's training. Note that in fact it impacts the convergence speed more than the convergence
> > itself, which is a relatively standard result. A degenerative effect of a higher learning rate starts to be visible
> > from $5 \times 10^{-2}$.
> >
> > | Learning Rate  | Maximum Accuracy after 100 epochs (\%) |
> > |----------------|-----------------------|
> > | $1 \times 10^{-4}$ |      39.57            |
> > | $5 \times 10^{-4}$ |     39.98             |
> > | $1 \times 10^{-3}$ |       40.43           |
> > | $5 \times 10^{-3}$ |       40.32          |
> > | $1 \times 10^{-2}$ |        40.34            |
> > | $5 \times 10^{-2}$ |         38.5              |

---

> > > ### Author Response · Authors · 2024-11-25
> > > **Theoretical Analysis and additional experiments**
> > >
> > > We propose here to adapt the idea of the reviewer and to compare alternative learning rules to the one we proposed.
> > >
> > > Let us remind the notations:
> > > - $ B $ is the feedback matrix.
> > > - $ \eta $ is the learning rate.
> > > - $ \cos(w) $ is the cosine similarity between $ B $ and $ g_l $, computed as:
> > >   $
> > >   \cos(w) = \frac{B \cdot g_l}{\|B\| \|g_l\|}.
> > >   $
> > > - $ \tilde{g} $ is the gradient $ g_l $, rescaled to match the norm of $ B $:
> > >   $
> > >   \tilde{g} = g_l \cdot \frac{\|B\|}{\|g_l\|}.
> > >   $
> > > to ensure the added component has a constant impact on the direction update of the feedback matrix.
> > >
> > >
> > > We consider two alternatives:
> > > - $B \leftarrow B - \eta (1 - \cos(w)) B + \eta \cos(w) \tilde{g},$ denoted as Alternative 1
> > > - $B \leftarrow (1 - \eta ) B + \eta  \tilde{g},$ denoted as Alternative 2
> > >
> > > We illustrate Figure 5 (in the Appendix), the original update when compared to the two proposed alternatives.
> > > The setup of Figure 5 is the following: $g_l$ is fixed and we compare the impact of the different learning rule on the cosine similarity of $B$ with $g_l$. The matrices dimensions are (200x200). The mean across 10 random initialisations of $B$ and $g_l$ is represented.
> > >
> > > It shows that **the original rule only performs marginally worse than the alternative 1**, but follows the same trend, while alternative 2 is definitely worse than the two others, with final cosine similarity (after 50 training steps) of 0.
> > > **We will add experiments using alternative 1 to the final version of the paper.**
> > >
> > > ---
> > > Alternative 1 further allows us to derive theoretical results in an ideal and easy setting:
> > >
> > > Let us rewrite
> > > $B = B_{\parallel} + B_{\perp},$
> > > where:
> > > - $ B_{\parallel} = \frac{(B \cdot g_l)}{\|g_l\|^2} g_l $ is the parallel component of $B$ to $g_l$
> > > - $ B_{\perp} = B - B_{\parallel} $ is the orthogonal component of $B$ to $g_l$
> > >
> > >
> > > The update rule for the parallel component is:
> > > $
> > > B_{\parallel}^{t+1} = B_{\parallel}^t - \eta (1 - \cos(w)) B_{\parallel}^t + \eta \cos(w) \tilde{g}.
> > > $
> > > Which can read:
> > > $
> > > B_{\parallel}^{t+1} = (1 - \eta (1 - \cos(w))) B_{\parallel}^t + \eta \cos(w)  \tilde{g}.
> > > $
> > >
> > > The update rule for the orthogonal component is:
> > > $
> > > B_{\perp}^{t+1} = B_{\perp}^t - \eta (1 - \cos(w)) B_{\perp}^t.
> > > $
> > > Since $ \cos(w) \leq 1 $, the factor $1 - \eta (1 - \cos(w)) $ reduces $ B_{\perp} $ with each step, causing it to decay exponentially:
> > > $$
> > > B_{\perp}^{t+1} = (1 - \eta (1 - \cos(w))) B_{\perp}^t.
> > > $$
> > >
> > > We can write:
> > > $
> > > B^{t+1} = B_{\parallel}^{t+1} + B_{\perp}^{t+1}.
> > > $
> > > As $ t \to \infty $:
> > > - $ B_{\perp} \to 0 $, because the orthogonal component decays exponentially.
> > > - $ B_{\parallel} $ aligns with $\tilde{g}$, since:
> > >   $$
> > >   B_{\parallel}^{t+1} \to \frac{B \cdot g_l}{\|g_l\|^2} g_l.
> > >   $$
> > >
> > > Thus, $B \to \tilde{g}$, achieving full alignment.
> > >
> > > ---
> > > As a conclusion, alternative 1 inspired by the reviewer remarks ensures:
> > > 1. Exponential decay of orthogonal components.
> > > 2. Preservation and alignment of parallel components.
> > > 3. Smooth and stable convergence to $\tilde{g}$, meaning a total alignment with $g_l$.
> > >
> > > However as illustrated Figure 5, it only performs marginally better than the original learning rule that does not offers such guarantees. We will test and compare on the paper's experimental setup if adding a gradient component enhances the original rule's performance.

---

> > > > ### Comment · Reviewer_xxPy · 2024-11-25
> > > > **> Theoretical Analysis and additional experiments**
> > > >
> > > > Thank you for the additional theoretical analysis.
> > > > However, it is quite surprising, as it seemingly shows that without my proposed modification, the $\tilde g$ term, the paper's update rule should cause B "to decay exponentially", since there would not be a term removing the effect of the collapse like here. Note also that since $B$ changes with time, the value of the cosine should also change with time (although this does not really changes the point of the authors). How do the authors explain this possibility?
> > > >
> > > > The result in Figure 5 is interesting, but also raises several questions. Why does the matrix converge to a value not aligned fully with the gradient (when using Alternative 1)? How is the alignment as good (almost) with the standard update rule ?

---

> > > > > ### Author Response · Authors · 2024-11-25
> > > > > **Clarification about the theoretical analysis**
> > > > >
> > > > > ### Theoretical analysis
> > > > > We do not agree with your point. In the original learning rule case, the orthogonal part of the matrix will indeed collapse and decay to zero. However, the other part of the matrix will become completely parallel to the gradient. This is notably due to the part that we renormalize the feedback matrix at each step to keep the matrix norm constant.
> > > > >
> > > > > In this case at the end, supposing the ideal case where we both have the full jacobian matrix and enough steps to apply the original learning rule, the matrix $B$ would converge towards a matrix parallel to the jacobian but where some directions (those that were orthogonal to the jacobian) are 0. It would thus have a positive cosine similarity, but not 1 as the parts of the initial $B$  orthogonal to the jacobian would have decayed to 0.
> > > > >
> > > > > ### About Figure 5
> > > > > We use an initial learning rate of $0.1$ with a scheduler of 0.99 each step to better mimic what happens in the experiments we ran (and are running to test alternative 1) which would explain the convergence towards a value lower than 1.
> > > > >
> > > > > If we use a learning rate of 1, alternative 1 and 2 converges towards 1. However, in real settings we do not have the full jacobian, but only a jacobian product estimate and we do not want all the rows of the feedback matrix to have a cosine similarity of 1 with a row vector at each step of the training. Thus we use smaller learning rate and a scheduler. Here the learning rate used can be higher as we are in a simpler setting, where the gradient we aim to align with is a fixed matrix.
> > > > >
> > > > > With different usable initial learning rates we systematically find that the initial learning rule performs relatively well compared to the alternatives, except for the very high learning rate where the alternatives converge to a value close to 1 while the original no.
> > > > >
> > > > > Zoutendijk's theorem states that convergence is guaranteed if the cosine similarity with the true gradient is strictly positive, which is the case with the original rule. Although the refinement you propose might lead to further improvement and the experiments will be added to the final version of the paper, our method provides a convergence guarantee for parallel feedback method applied to a wide variety of networks.

---

> > > ### Comment · Reviewer_xxPy · 2024-11-25
> > > **> Learning rule explanation**
> > >
> > > I think the authors misunderstood my point, I mean that the update rule itself, as presented, is only a rescaling, a scalar multiplied by the feedback matrices. However, something must be unclear because I do not get how a rescaling and a renormalization would modify the matrices. What do the authors mean by "by row"? This is something that is not explained in the article, and I feel like this is necessary to finally get an understanding of the learning rule proposed. Can you clarify the dimensions of the different elements considered notably?
> > >
> > > Thank you for the Figure 4. However the caption is quite hard to read and there are no labels to the legend, making it very unclear. Why is there no convergence without noise? Without understanding the value in the legend I can't really understand the figure.
> > >
> > > Thank you for the Table. Since (from what is claimed) the main importance of the authors' learning rule is to collapse unwanted dimensions, this result seems logical, thank you.

---

> > > > ### Author Response · Authors · 2024-11-25
> > > > **Clarification about the learning rule**
> > > >
> > > > First of all, we would like to thank you for your implication and the time you spend in this reviewing process.
> > > > We will try to answer every further questions to the best of what we can until the process stops.
> > > >
> > > > ### About the learning rule:
> > > > There seems to be a misunderstanding, caused by an unclear notation of the update rule equation. Here is a clarification of it and we will make sure to change accordingly in the final version of the paper:
> > > > We wrote: $B \leftarrow B - \eta (1 - \cos(w)) B$, however it is unclear.
> > > >
> > > > The estimation of the gradient obtained via forward auto-differentiation is basically calculated thanks to jacobian vector product according to one perturbation vector (called the tangent).
> > > > The estimation $g_l$ is thus **a row vector**.
> > > > The cosine similarity $cos(w)$ is in fact the **vector column** of which each entry $cos(w)_i$ is the cosine similarity of the row vector $g_l$ with the $i$-th row of $B$. **It is thus a vector and not a scalar**. To clarify this, we will denote it $\textbf{cos(w)}$.
> > > > What we denote by 1 should actually be denoted $\textbf{1}$ as it is a vector of which each entry is a 1.
> > > >
> > > > The simple product between the vector $(\textbf{1}-\cos(w))$ and $B$ is actually an **Hadamard (or element-wise) product**, denoted $\circ$.
> > > >
> > > > The correct notation is thus:
> > > > $$B \leftarrow B - \eta (\textbf{1} - \textbf{cos(w)}) \circ B$$
> > > >
> > > > We hope this clarify the misunderstanding on how the learning rule operates.
> > > >
> > > > ### About Figure 4
> > > > We will redo the figure so that the caption is easier to read.
> > > >
> > > > Every curve indicate a different feedback matrix. The x-axis indicates the number of steps used to train using the original learning rule, while the y-axis indicates the mean of the vector $\textbf{cos(w)}$.
> > > > They were chosen to have different initial cosine similarity with the fixed gradient matrix to illustrate how the learning rule operates regarding different scenario (mean cosine similarity ranging from $\[-0.9, -0.3, 0, 0.3, 0.9\]$.
> > > > We also showed that the naive learning rule does not manage to improve the cosine similarity if it is exactly orthogonal (red curve stays at 0).
> > > > However, adding small noise helps get a positive cosine similarity (cyan curve).
> > > > We hope this clarifies what we wanted to illustrate in this Figure. We will make sure to make it more readable and to explain it in the final version of the paper.

---

> > > > > ### Comment · Reviewer_xxPy · 2024-11-26
> > > > >
> > > > > **About the learning rule**
> > > > >
> > > > > Thank you for your answer, it is indeed helping to produce a slightly clearer update rule. I'm sorry to continue on this point, but the dimensionality of the different quantities is however still confusing me a lot. Let us consider $m=n_{l-1}$ and $n=n_l$ for easiness here, and $o$ the size of the output $y$ and of the error $e$. The layer weight $W_l$, is thus of size $m \times n$ and the matrix $B_l$ of size $n \times o$. First, the equation (4) seems false, rather than $\frac{∂L(x)}{∂h_i }U_i$, it should probably be $\frac{∂L(x)}{∂h^i_l }U^i_l$. The same errors are present in the previous equation on weight-perturbated forward gradients. Here, we have the matrix Jacobian of size $m$ (the size of the input), multiplied by a (rescaled) random perturbation of size $n$. Thus, the size of $g_{a,l}$ is a matrix of size $m \times n$, like the weights, which is logical as the gradient approximation used to modify the weights. The equation of $cos(\bar w_l)$ is then indeed surprising, especially as equation (5) compares it to a scalar. If we consider that it is a row vector, by comparing each column/row of $g_{a,l}$ to $B_l$, which is of size $n \times o$, then we obtain that $cos(\bar w_l)$ is of size $m \times o$. Then, it is not possible to do an element-wise product between $cos$ and $B_l$. It could be possible to compute a product between them to obtain a matrix of size $n \times m$, but this is not the same size as $B_l$ of which it is deduced.  Where did I go wrong in this computation?
> > > > >
> > > > > I'm sorry to take a lot of time on this point, but I feel like making this update rule crystal clear is necessary for the authors to sell their method, which does seem of interest.
> > > > >
> > > > > **Figure 4**
> > > > >
> > > > > Thank you. However, I am (once again) a bit confused by the quantities being considered here. I guess that the gradient of size (4x4) is the equivalent of $g_{a,l}$. What is the dimension of $B_l$ here? What if the matrices were not squares? If the gradient matrix is fixed, how is random noise enough to obtain a positive alignment, and where is the noise added in the equation? Why are the matrices with negative alignment obtaining better final cosine similarity compared to the positive equivalent? And I must note again that showing no alignment for the standard update rule without noise is concerning, in particular when noise is not noted in the article and seems absolutely necessary here.
> > > > >
> > > > > **Theoretical analysis and Figure 5**
> > > > > I should probably wait to get a better understanding of the learning rule before commenting here.

---

> > > > > > ### Comment · Reviewer_R1hf · 2024-11-26
> > > > > > **Looking forward to these answer**
> > > > > >
> > > > > > Reviewer xxPy summarizes many of my concerns with regards to the matrix dimensions and how the rescaling affects training. I am looking forward to the authors response too. I strongly appreciate the effort of the authors in explaining their algorithm in great details.

---

> > > > > > > ### Author Response · Authors · 2024-11-28
> > > > > > > **Notations and Discussion**
> > > > > > >
> > > > > > > We would first like to thank the reviewers for their time and patience and constructive feedbacks to understand our proposition.
> > > > > > > We believe this strengthens and clarifies the proposed learning rule.
> > > > > > > Every correction and suggestion will be taken into account in the final version of the paper.
> > > > > > >
> > > > > > > We acknowledge a lack of clarity in Section 3.2 that we will update in the next version of the paper.
> > > > > > > This is due to the fact that we indisctinctively used in the paper (and some of our answers) the term "gradient" to refer to the gradient of the loss with respect to the pre-activation of the
> > > > > > > layer $l$, $\nabla \mathcal L_{l}\equiv \frac{\partial \mathcal{L}}{\partial a_l}$ and the gradient of the output with respect to the pre-activation of the
> > > > > > > layer $l$, which is in fact the **Jacobian** of the layer: $J_{l}\equiv \frac{\partial \hat{y}}{\partial a_l}$.
> > > > > > > We will provide here a detailed
> > > > > > > explanation of the role of forward gradients in the update of the feedback matrices.
> > > > > > > We will make sure the following explanation replaces the current one in the final version of the paper.
> > > > > > >
> > > > > > > ### Notations clarifications
> > > > > > >
> > > > > > > L.096 does not show how the Jacobian of the layer is used to convey the gradient of the loss with respect to the output of the network.
> > > > > > > Specifically, we write $\nabla \mathcal L_{l}\equiv \frac{\partial \mathcal{L}}{\partial a_l}$  as the gradient of the loss with respect to the pre-activation of a specific layer $l$.
> > > > > > >
> > > > > > > We should also specify that the error $e=\frac{\partial \mathcal{L}}{\partial a_L}$ is conveyed to update the weights of layer $l$ via backpropagation through the Jacobian of the layer $l$ and
> > > > > > > add the following to equation (1):
> > > > > > > >$$\delta a_l = (J_{l}^T e) \odot\sigma_l'(a_l) $$
> > > > > > > When comparing this update to equation (2),
> > > > > > > > $$\delta a_l = (B_l e) \odot \sigma_l'(a_l)$$
> > > > > > >
> > > > > > > it becomes clearer that what we want to align the feedback matrices to are the transpose of the jacobians of the
> > > > > > > corresponding layers.
> > > > > > >
> > > > > > > Equivalently, equation (3) measures the cosine similarity, i.e the alignment between corresponding rows of the Jacobian transpose $ \mathbf{J}_l^\top $ and the feedback matrix $ \mathbf{B}_l $. It is defined as:
> > > > > > >
> > > > > > > >$$
> > > > > > > \cos(\omega_l) = \frac{1}{n} \sum_{i=1}^n \frac{ (\mathbf{J}_l^\top[i])^\top \cdot \mathbf{B}_l[i] }{ \| (\mathbf{J}_l^\top[i])^\top \| \| \mathbf{B}_l[i] \| }
> > > > > > > $$
> > > > > > >
> > > > > > > **Where:**
> > > > > > >
> > > > > > > - $ \mathbf{J}_l^\top[i] $ is the $ i $-th row of $ \mathbf{J}_l^\top $ (thus a column of $\mathbf{J_l}$).
> > > > > > > - $ \mathbf{B}_l[i] $ is the $ i $-th row of $ \mathbf{B}_l $.
> > > > > > >
> > > > > > >
> > > > > > > Lastly, line 279: we will correct the equation to:
> > > > > > > $$
> > > > > > > \cos(\bar{\omega_l}) = \frac{1}{n} \sum_{i=1}^n \frac{ (\mathbf{\tilde{J}}_l^\top[i])^\top \cdot \mathbf{B}_l[i] }{ \| (\mathbf{\tilde{J}}_l^\top[i])^\top \| \| \mathbf{B}_l[i] \| },
> > > > > > > $$
> > > > > > > where $\mathbf{\tilde{J_l}}$ is the estimation of $\mathbf{J_l}$ via Jacobian vector product (either through weights or activations perturbations).
> > > > > > >
> > > > > > > ---
> > > > > > > ### Weight Perturbed vs. Activation Perturbed Jacobian Vector Product
> > > > > > >
> > > > > > > Section 2.2 discusses how to evaluate with the least possible variance an estimate of $\nabla \mathcal L_{l}$ via Jacobian vector product.
> > > > > > >
> > > > > > > Reviewer xxPy is right to point out that there are notation inconsistencies to define the two alternatives to perturb the layer $l$. We will make sure to correct this error in the paper.
> > > > > > >
> > > > > > > We would also like to clarify that the two alternatives are not equivalent and as it was pointed out by reviewer R1hf, the paragraph
> > > > > > > lines 231-237 should be modified to take into account that in earlier layers of convolutional networks, the number of parameters is smaller than the number of neurons, meaning the sentence
> > > > > > > > "As the number of neurons n is usually considerably lower than the number of weights"
> > > > > > >
> > > > > > > will be modified to take into account this remark. Evaluating the best perturbation space is left for future work and we focus here on activation perturbations.

---

> > > > > > > > ### Author Response · Authors · 2024-11-28
> > > > > > > > **Shapes study**
> > > > > > > >
> > > > > > > > Let us now study the shapes of the considered quantities:
> > > > > > > >
> > > > > > > > Let us consider $m=n_{l-1}$ and $n=n_l$ the shapes of the input and output of the layer $l$.
> > > > > > > >
> > > > > > > > Let us denote $o$ the size of the output $\hat{y}$ of the network.
> > > > > > > >
> > > > > > > > The weight matrix $W_l$ of the layer $l$ is of shape $(n \times m)$, the error $e$ is of shape $(o \times 1)$ and the Jacobian $J_l$ is of shape $(o \times n)$.
> > > > > > > >
> > > > > > > > The feedback matrix $B_l$ is of shape $(n \times o)$.
> > > > > > > > - When perturbing the weights, we apply a perturbation $\delta w_l$ to the weights $W_l$. This is outputed by the layer as $\delta w_l \cdot h_{l-1}$, where $h_{l-1}$ is the input of the layer $l$.
> > > > > > > > It results in a perturbation vector on the Jacobian of shape $(n \times m) \times (m \times 1) = (n \times 1)$.
> > > > > > > >
> > > > > > > > - When perturbing the activations, we apply a perturbation $\delta a_l$ to the pre-activations $a_l$. This is outputed by the network as $\sigma_l'(a_l) \odot \delta a_l$, where $\sigma_l'(a_l)$ is the derivative of the activation function of the layer $l$.
> > > > > > > > It results in a perturbation vector on the Jacobian of shape $(n \times 1)$ as the Hadamard product involves two vectors of size $(n \times 1)$.
> > > > > > > >
> > > > > > > > No matter the considered perturbation, let us denote $\mathbf{\delta} $ the perturbation vector of shape $(n \times 1)$ applied to the Jacobian $J_l$ of shape $(o \times n)$.
> > > > > > > >
> > > > > > > > Directly aligning the estimated Jacobian with the feedback matrix  along $\mathbf{\delta}$ would require aligning
> > > > > > > > $ J_l \mathbf{\delta}$ with $B_l^\top \mathbf{\delta}$, which we in fact do by aligning $\tilde{J_l}^\top$ with $B_l$, where:
> > > > > > > >
> > > > > > > > $$
> > > > > > > > \tilde{J_l} = J_l \mathbf{\delta}\mathbf{\delta}^\top
> > > > > > > > $$
> > > > > > > >
> > > > > > > > The resulting matrix is of shape $(o \times n)$ and we use the corrected equation 6 to compute the cosine similarity between the feedback matrix $B_l$ and the estimated jacobian transposed.
> > > > > > > >
> > > > > > > > In the example discussed in a previous comment, we mentioned
> > > > > > > > > "The estimation of the gradient obtained via forward auto-differentiation is basically calculated thanks to jacobian vector product according to one perturbation vector (called the tangent). The estimation is thus a row vector."
> > > > > > > >
> > > > > > > > What we meant is that it is actually a rank 1 matrix because it is the outer product of two vectors. As an example:
> > > > > > > >
> > > > > > > > Let us consider $\delta$ as a one-hot vector. For example, such that $ \delta = [1, 0, 0, \dots, 0]^T $.
> > > > > > > >
> > > > > > > > The product $ J \cdot \delta $ isolates the first column of $ J $, and
> > > > > > > > $
> > > > > > > > J \cdot \delta =
> > > > > > > > \begin{bmatrix}
> > > > > > > > J_{11} \\
> > > > > > > > J_{21} \\
> > > > > > > > \vdots \\
> > > > > > > > J_{m1}
> > > > > > > > \end{bmatrix} \in \mathbb{R}^m.
> > > > > > > > $
> > > > > > > > This corresponds to the first column of $ J $.
> > > > > > > > The outer product $ (J \cdot \delta) \cdot \delta^T $ places the first column of $ J $ into the **first column of the output matrix**, with all other columns being zero. This isolates the **first column of $ J $**, while all other columns are zero.
> > > > > > > >
> > > > > > > > These rank one matrices are summed over time to get a more precise approximation of the jacobian to align $B_l$ with.

---

> > > > > > > > > ### Comment · Reviewer_xxPy · 2024-11-30
> > > > > > > > >
> > > > > > > > > Thank you.
> > > > > > > > > However, I'm becoming even more confused, for several reasons. The authors had previously written that "cos in fact a vector column". Now, the equation written by the authors of $cos(\bar w_l)$ is once again a scalar value, summing the cosine similarity of each individual row (of size $o$) of the pseudo-Jacobian with the associated row of the backward matrix (so each one a scalar value).
> > > > > > > > > The authors have also changed the quantity computed with forward-mode AD. Now, it is the directional derivative of a jacobian with output size $o$.  This seems indeed like the good quantity to align with for $B$, but this is again a late change...
> > > > > > > > > "These rank one matrices are summed over time to get a more precise approximation of the jacobian to align with": I don't know that the authors are referring to in the article.

---

> > > > > > > > > > ### Author Response · Authors · 2024-12-01
> > > > > > > > > >
> > > > > > > > > > ###  Cosine Similarity Notation and Explanation:
> > > > > > > > > > You are correct that the earlier notations and explanations for cosine similarity were not sufficiently rigorous, which contributed to confusion.
> > > > > > > > > > The cosine similarity averaged to get a scalar value is indeed very misleading. This averaged measure is what indicates alignment of the update path with the true gradient direction, ensuring convergence if strictly positive thanks to Zoutendijk's theorem.
> > > > > > > > > >
> > > > > > > > > > In the update rule every $cos(\tilde{w}_l)_i$ is used to update the corresponding $B_l[i]$, thus the sentence "cos in fact a vector column".
> > > > > > > > > >
> > > > > > > > > > In the presented explanation in the previous comment, $cos(\tilde{w_l})$ is clarified as a scalar value obtained by summing the cosine similarities for each individual row. This is to help link it with Zoutendijk's theorem but should be made clearer. We thank you for your thorough examination of our answer as it could effectively be misleading to adopt **non strictly rigorous** notations as it was the case and we apologize for it.
> > > > > > > > > >
> > > > > > > > > > There should indeed thus be a clarification in the learning rule,  using $cos(\tilde{w}_l)_i$ for every corresponding row of every $B_l[i]$, and the scalar value $cos(\tilde{w_l})$ in equation 5 to ensure convergence.
> > > > > > > > > >
> > > > > > > > > >   ###  Forward-Mode Automatic Differentiation (AD):
> > > > > > > > > > It is important to emphasize that forward-mode AD has always been implemented in the described manner. The directional derivative of the Jacobian with output size was consistently the target quantity for alignment with B. The recent explanation simply provides a more rigorous and precise articulation of this, without introducing any changes to the underlying methodology.
> > > > > > > > > >
> > > > > > > > > > ###    Gradient Averaging for Mitigation:
> > > > > > > > > > As stated in lines 321/322 of the original paper, the cumulative moving average is used to mitigate issues during training.
> > > > > > > > > >     This process helps refine the approximation of the Jacobian over time, ensuring a more stable and precise alignment, and is a fundamental aspect of the described methodology.
> > > > > > > > > >
> > > > > > > > > > We hope this addresses your concerns and provides clarity on these aspects. Thank you again for your engagement and patience, we are eager to continue this discussion if more clarifications are needed.

---

> > > > > > > > > > > ### Comment · Reviewer_xxPy · 2024-12-02
> > > > > > > > > > >
> > > > > > > > > > > The authors have clarified that the cosine value used in the actual learning rule is therefore a vector of size $n$. This (finally?) gives the learning rule used (without tackling the noise). Therefore the mechanism seems to, rather than truly "align" the feedback matrix $B$, to collapse (following their theoretical analysis) unwanted columns of the feedback matrix $B$ with poor alignment. This is quite different in my opinion.
> > > > > > > > > > >
> > > > > > > > > > > Indeed, I missed these lines, thank you. Once again, this is a major algorithmic choice that needs to be clarified at least in Algorithm 1.
> > > > > > > > > > >
> > > > > > > > > > > I thank the authors for the in-depth discussion on their learning rule, which I hope will help future submissions of this work.
> > > > > > > > > > > Still, I will remain on a rating of 3 for the following reasons.
> > > > > > > > > > > - There are too many major changes that are necessary in the manuscript to even tackle the discussion we had.
> > > > > > > > > > > - The learning rule proposed must be rewritten entirely and clarified.
> > > > > > > > > > > - The paper requires *consistent* and *clear* notations, as many mistakes are seemingly still present. All elements like the use of noise or the moving average must be added or clarified.
> > > > > > > > > > > - A clear justification for the learning rule is still missing, although the theoretical analysis provided by the authors is a welcome start.
> > > > > > > > > > > - The subsequent ablations point to several major issues: non-complete alignment to the gradient, compared to the claims of the authors. High dependency on the value of the gradient aligned: the learning rule does not work without noise in the (average) case of poor alignment.
> > > > > > > > > > > - The link with Zoutendijk's theorem is now very unclear as both the cosine values considered are 'by row', and the alignment of the actual gradient used to update the parameters (not the forward gradient) is not provided.
> > > > > > > > > > >
> > > > > > > > > > > I feel like the idea of the authors is worth following and seems promising seeing the accuracies provided, but that this submission requires a very thorough rework before being submitted again. Once again, I thank the authors for this discussion which I feel was productive for their work.

---

### Official Review · Reviewer_JB9x · 2024-11-01

**Soundness:** 3
**Presentation:** 2
**Contribution:** 3
**Rating:** 5
**Confidence:** 4

**Summary:**

The authors present Gradient Aligned Projected Error (GrAPE), a backpropagation-free learning algorithm aimed at parallelizing the training of deep neural networks by aligning feedback paths with forward gradients. GrAPE, which modifies the Direct Feedback Alignment (DFA) approach, is evaluated on tasks requiring modern deep architectures such as AlexNet, VGG-16, and Transformer-based models. The results show significant improvements over other backpropagation-free methods, especially in scaling to complex architectures, although it still lags behind backpropagation in generalization performance. Notable is its extension to convolutional layers, which other feedback alignment methods have lacked so far.
Their modification of DFA includes aligning feedback matrices with forward gradients, which they do with the help of the Zoutendijk theorem. They proved that the method converges and has good empirical results in their experiments. The authors highlight GrAPE’s biological plausibility and its potential for future neuromorphic applications.

**Strengths:**

i) The authors propose a new update rule for the direct feedback matrices to align their directions with the corresponding gradients. The goal of this update is thus to minimize the orthogonal directions while improving the gradient alignment. This is about the analysis with Feedback Alignment algorithms, which require the angle between the feedback and the true gradient of the forward process to be less than 90 degrees.

ii) Update ensures that the parallel directions with the gradients will be promoted for error projection.

iii) The method successfully addresses the limitations of DFA with respect to convolutional layers.

iv) The experimental evaluation is extensive, including performance on both shallow and deep architectures across datasets (MNIST, CIFAR-10, CIFAR-100, WikiText-103). The results clearly demonstrate that GrAPE outperforms DFA, especially in architectures with complex structures like transformers and VGG-16.

v) The approach is grounded in the Zoutendijk theorem for optimization, which provides a convergence guarantee when feedback directions align with gradient directions. This theoretical underpinning gives the algorithm a sound basis for further exploration.

**Weaknesses:**

1) The Zoutendijk theorem could have been better explained with and without context to the DFA issue. The readers have to understand the theorem to relate to the claims based on it.

2) The presentation could have been better.

3)  The paper mentions a need for further research on regularization techniques, as the authors have noted.

4) While the algorithm addresses some parallelization issues, the paper does not elaborate on the computational overhead and variance in forward gradient estimations.



.

**Questions:**

1) How much degradation does it come with for ImageNet? I understand that the process is still in the development stage, this question is not to penalize the paper contributions.

2) How much difference in accuracies are present when the conv layers are implemented with and without GrAPE (with rest of FC layers implemented with GrAPE)?

3) There are methods AugLocal (ICLR 2024) and InfoPro (ICLR 2021) that explain why the feedback alignment methods are failing for deeper networks. In that case, the linear separability of features must be addressed. This property shows how much the earlier layers in the network are neglecting the subsequent layers in adapting to the local loss objective. Could that analysis be done for this method? An interesting understanding might emerge.

4) Could you provide the computational overhead required in forward gradient estimations?

---

> ### Author Response · Authors · 2024-11-18
> **Answer to weaknesses 1 - 4**
>
> ## 1. **Clarifying the Zoutendijk Theorem in the Context of DFA**
>
> We appreciate the reviewer’s feedback regarding the Zoutendijk theorem. This theorem ensures global convergence in optimization when the search direction maintains an angle with the steepest descent direction that is bounded away from $ \pm \frac{\pi}{2} $. In our context, this theorem is crucial for DFA-based methods, as it highlights the importance of aligning feedback matrices with the true gradient direction. If the feedback matrices were orthogonal to the gradient direction, convergence could not be ensured, leading to instability or poor performance. Random feedback matrices, which DFA often relies on, can misalign with the gradient, presenting a significant challenge.
>
> In GrAPE, we address this by updating the feedback matrices to achieve consistent alignment with the forward gradient direction. This helps ensure that the feedback matrices do not drift into directions orthogonal to the gradient, satisfying the convergence conditions outlined by the Zoutendijk theorem. We will clarify this explanation in the paper to make the relevance of the Zoutendijk theorem more accessible, providing readers with a clearer understanding of its application in GrAPE.
>
> ## 2. **Suggestions for Improved Presentation**
>
> We acknowledge the reviewer's suggestion to enhance the presentation, particularly around technical explanations like the Zoutendijk theorem. In response, we plan to restructure sections where theoretical foundations are introduced, adding more detailed background explanations. If the reviewer has other specific suggestions to improve the clarity and accessibility of the paper, we would be grateful for any additional guidance.
>
> ## 3. **Regularization Techniques and Further Research**
>
> The reviewer’s suggestion to explore regularization techniques is well taken. While regularization strategies like dropout, batch normalization, and weight decay are standard in backpropagation, they may require adaptation for feedback alignment methods like GrAPE. We noted this as a potential area for further research, as GrAPE’s generalization gap, particularly on larger datasets, could benefit from refined and dedicated regularization.
>
> ## 4. **Parallelization and Computational Overhead of Forward Gradients**
>
> GrAPE effectively addresses some of the parallelization challenges inherent in BP. However, it is true that calculating forward gradients incurs additional computational overhead. In our implementation, we observed an approximate 15-20\% increase in computation time per epoch compared to DFA, though this may vary with network depth, hardware, and batch size. The current implementation also relies on the one done in [Biotorch](https://github.com/jsalbert/biotorch/tree/main) which relies on backward hooks to efficiently modify the "gradient". (see [implementation example of the linear layer](https://github.com/jsalbert/biotorch/blob/main/biotorch/layers/dfa/linear.py) for more details).

---

> ### Author Response · Authors · 2024-11-18
> **Answers to Specific Questions**
>
> **Q1: Degradation on ImageNet**
>
> Thank you for this question, as it emphasizes the importance of evaluating GrAPE’s limitations on larger, more complex datasets. While we haven’t yet conducted extensive testing on ImageNet, the experiments with Transformers on WikiText-103, shows that GrAPE can be used on datasets of over 100 million tokens.
>
> **Q2: Accuracy Differences with GrAPE in Convolutional Layers**
>
> For convolutional layers, GrAPE demonstrates a noticeable performance improvement over DFA, aligning feedback more closely with the gradient direction. This improvement suggests that GrAPE effectively leverages structured feedback, which is particularly beneficial for convolutional filters.
>
> This observation connects interestingly with principles from Extreme Learning Machines (ELM). In ELM, hidden layer weights are initialized randomly and left unchanged, with only the output layer being trained with BP. In this sense, while the feature extractor is fixed in ELM settings, it is updated in a random fixed direction in DFA setting and in a learned approximate of the true gradient direction in GrAPE setting.
> We plan on including the proposed comparison on AlexNet to showcase potentially interesting dynamics.
>
> **Q3: Relation to AugLocal and InfoPro**
>
> Thank you for suggesting AugLocal (ICLR 2024) and InfoPro (ICLR 2021). Both methods highlight challenges in feedback alignment due to difficulties in ensuring linear separability in early layers, potentially causing feedback signals that don’t support optimization goals in subsequent layers.
>
> By enhancing feature separability (AugLocal) and preserving information quality across layers (InfoPro), these methods provide useful strategies that could complement or improve feedback alignment mechanisms like GrAPE. However, it would require introducing local augmentations and regularization techniques which would hinder the interpretability of the presented results.
>
> GrAPE addresses the linear separability issue by aligning feedback signals with the forward gradient, which promotes directional consistency with the layer-wise gradient. Analyzing GrAPE’s effect on linear separability could indeed offer valuable insights and is indeed a promising avenue for future research, going however beyond the scope of the current work by introducing local losses.
>
> **Q4: Computational Overhead of Forward Gradient Estimations**
>
> We answered to this specific question when discussing weakness 4.
>
> ---
>
> We hope these clarifications address the reviewer’s concerns and we are open to further feedback and discussion with the reviewer to enhance the quality and presentation of our work.

---

> > ### Comment · Reviewer_JB9x · 2024-11-28
> >
> > I thank the authors for providing an extensive explanation of the queries on the paper.
> > I would be keeping my score.

---

### Official Review · Reviewer_R1hf · 2024-11-05

**Soundness:** 3
**Presentation:** 2
**Contribution:** 2
**Rating:** 3
**Confidence:** 4

**Summary:**

The presented paper proposes an Adaptive Direct Feedback Alignment method which mixes together Forward Gradient calculation and Direct Feedback Alignment. The underlying idea is to augment standard DFA algorithm with forward gradient calculation, in order to derive the cosine angle between the feedback matrix and the true gradient. The authors uses this extra information to make their algorithm adaptive, by updating the feedback matrices before projecting the error. This favors a better alignment with the projected error and the true gradient, yielding much improvements over backpropagation-free alternatives while still lagging behind backpropagation.

**Strengths:**

1. The paper is generally well-written, albeit with some (very concerning) exceptions. The introduction, background and related work are very understandable, which is not often the case in the dedicated literature.
2. The experiments clearly show results in favor of the proposed method against backpropagation-free alternative. More precisely, authors obtains figure on par or better than standard FA, which itself surpasses standard DFA. Thus, the authors are able to improve the gap with backpropagation while keeping their algorithm parallelizable.
3. The authors acknowledge and characterize a stronger overfitting problem than standard DFA.
4. The variety of architectures studied in the experiments is relevant with respect to the state-of-the-art.

**Weaknesses:**

**Crucial weaknesses:**
1. There is a simple but major problem with equation (6). The matrices $B_l^t$ does not depend on the input $x$, but the cosine angle does. The cosine angle is defined as
$$
\cos(\omega_l) = \frac{g_{a,l}^{\top} B_l}{||g_{a,l}|| \cdot || B_l ||}
$$
But the gradient $g_{a,l}$ depends on $x$. As a consequence, it is not clear how the algorithm is supposed to behave in a mini-batch setting. Accordingly, the pseudo-code in algorithm 1 is not clear since we iterate over the whole dataset before updating the weights. Either the reviewer does not understand the dimensions of the matrices involved, if there is a mini-batch dimension, either the pseudo-code does not correspond to the actual implementation able to deliver such figures in the experiment section.

2. Albeit empirical performances on various datasets and models are provided, there no ablation on any aspect of the proposed algorithm. The authors claims a good compatibility with Batch-normalization without giving evidence. The authors does not specify how they rescale the feedback matrices, nor if the normalization plays a crucial role.

3. The authors shows that the cosine similarity of the forward gradient estimate increase with the number of JVP calculation. This is more a debugging experiment than an ablation study as the theory clearly predicts it would happen. It would actually be very interesting to follow the cosine similarity between the projected error and the true gradient on real dataset throughout training, and compare the different methods.

4. To the reviewer, the most crucial aspect of such paper would be to ablate activity perturbation vs weight perturbation. Putting aside the fact that it is not clear how the computations happens for activity perturbation in a batch setting, it intuitively seems more natural to try to align a feedback matrix with the weight gradient rather than the activity gradient.

**Small weakness:**
Even if the paper focuses on improving model performances, it would be nice to have to some runtime comparison between different methods. Even though it would not reflect the parallelization potential given that experiments are performed on single GPU, it would give an idea of the practical overhead induced by forward gradient calculations.

**Minor remarks:**
1. Line 51: typo “the the feedback…”
2. Line 258: typo “feedback path with and the gradient” → is it "with" or "width" ?
3. Fig.1.e → GrAPE instead of GAEP.

**Questions:**

Please address the crucial weaknesses in details.
If possible, it would be appreciated to have some more information about the small weakness.

---

> ### Author Response · Authors · 2024-11-18
> **Paper motivation and answer to Weakness 1**
>
> First of all, we would like to thank the reviewer for their time and insightful comments. The detailed feedback and suggestions provided are highly appreciated as they will help us improve the quality and clarity of our work. Below, we address each of the reviewer's concerns and suggestions.
>
>
> ## Difference between Backpropagation and DFA
> The crucial equation change between backpropagation and DFA is the replacement of the **weight-dependent Jacobian**  $\frac{\partial \hat{y}}{\partial a_l}$ with a **fixed feedback matrix** $ B_l$:
>
> As written in equation (1) of the paper, backpropagation reads, when unrolling the chain rule:
>
> $$
> \delta a_l = \left( \frac{\partial \mathcal{L}}{\partial a_l} \right)\odot \sigma'(a_l) =
> \left( e \times \frac{\partial \hat{y}}{\partial a_l} \right) \odot \sigma'(a_l)
> $$ where $\frac{\partial \hat{y}}{\partial a_l}$ is the corresponding Jacobian matrix.
>
> In DFA, however, the feedback matrix $B_l$ is used instead of the Jacobian matrix to map the error signal onto the corresponding weight space:
> $$
> \delta a_l = \left( B_l \times e \right) \odot \sigma'(a_l)
> $$
>
> The core idea of GRAPE is to calculate the directional information of the Jacobian during the forward phase via Jacobian-vector products, to progressively align the feedback signal $B_le$ with the corresponding gradient, thereby ensuring convergence. Additionally, by averaging the Jacobian estimations over batches, GRAPE averages the high-variance gradient estimate over multiple samples, which reduces variance and enhances the efficiency of the jacobian estimation.
>
> ## 1. Clarification on Equation (6) and Mini-Batch Behavior
>
> The reviewer raised a valid concern about Equation (6) and how it behaves in a mini-batch setting, given that the cosine similarity term, $ \cos(w_l) $, depends on the input $ x $ while $ B_l $ does not. We clarify this as follows, first we motivate our approach in a clearer manner, before adressing the mini-batch behavior issue highlighted by the reviewer.
>
> - **Role of $ B_l $ as an Approximate Jacobian**: The feedback matrix in DFA $ B_l $ serves an analogous role to the Jacobian in backpropagation. In traditional backpropagation, the Jacobian matrix represents the exact partial derivatives needed to propagate gradients back through each layer. In DFA, this matrix is randomly initialized and
> fixed, serving as a surrogate for the true Jacobian, without being correlated.
> In GrAPE, we approximate this gradient flow using $ B_l $, by aligning it with the forward gradient calculated via Jacobian-vector products (JVPs).
>
> If $ B_l $ were to exactly match the true Jacobian of each layer, GrAPE would yield updates identical to backpropagation. However, maintaining an exact Jacobian would reintroduce backpropagation's sequential dependencies and computational overhead, undermining GrAPE’s parallelizability. Instead, we iteratively align $ B_l $ in the direction of the true gradient using cosine similarity, giving GrAPE a progressively improving approximation of the Jacobian that enhances alignment and stability while preserving computational efficiency.
>
> - **Batch-Averaged Cosine Similarity**: To handle the input dependency in mini-batch settings, we calculate a batch-averaged cosine similarity. Specifically, for each mini-batch, we compute the cosine similarity term $ \cos(w_l) $ for each sample, accumulate these values across the mini-batch, and then use the averaged cosine similarity to update $ B_l $ once per mini-batch. This batch-averaged approach ensures that the feedback matrix update reflects a consensus direction across the mini-batch, stabilizing the alignment with respect to gradient fluctuations among individual samples.
>
> - **Pseudo-Code and Actual Implementation**: We recognize that the pseudo-code in Algorithm 1 could benefit from clearer notation to reflect this batch-wise process. In the revised version, we will specify that the cosine similarity is accumulated over all samples in the mini-batch before updating the feedback matrices, ensuring the pseudo-code aligns with the implementation used in our experiments.

---

> ### Author Response · Authors · 2024-11-18
> **Answer to weaknesses 2 - 4**
>
> ## 2. Ablation Studies and Compatibility with Batch-Normalization
>
> We appreciate the suggestion to include ablations to better illustrate the effect of different components in our algorithm. We acknowledge that a thorough ablation study would provide valuable insights into GrAPE's performance and behavior. Together with the choice of initialization, optimization algorithm and learning rate and even architectural details, these ablations can help us understand the impact of each component on the method's convergence and stability.
> Every component used in GrAPE was initially designed for BP, so understanding how they interact with each other in the context of GrAPE could dramatically improve the results and our understanding of the proposed method.
> A comprehensive ablation study, covering every aspect of the method however, would be beyond the scope of this paper as it would require a significant amount of computational resources. We will however include a more detailed discussion on the following points:
>
> - **Feedback Matrix Rescaling**: As mentioned in the paper, we rescale the feedback matrices $ B_l $ to have a constant norm following each update, which prevents the alignment direction from collapsing due to scale variations. This was done to follow the initial design of DFA, where the feedback matrices are initialized with a fixed norm.
> We recognize that further details on this rescaling process should be provided, and we will add this explanation in the supplementary materials, together with an ablation study of this component.
>
> - **Batch-Normalization Compatibility**: Batch normalization is a natural fit for GrAPE’s design because the feedback matrices $ B_l $ are rescaled to maintain a constant norm after each update, ensuring that the error signal’s feedback remains stable in magnitude. To achieve consistency, it is beneficial to keep the forward signal similarly normalized, which batch normalization facilitates by maintaining activations at a controlled scale throughout training. This alignment in signal scaling helps GrAPE avoid discrepancies in feedback and forward pathways, stabilizing the training process and supporting effective feedback alignment.
> In the reported experiments, we thus used batch normalization in the other methods to ensure a fair comparison.
> We will include a more detailed discussion on the compatibility of GrAPE with batch normalization in the revised paper, highlighting the benefits of this combination for training stability and convergence.
>
>
> ## 3. Analysis of Cosine Similarity over Time
>
> The reviewer noted that our experiment showing the increase in cosine similarity with more JVP samples aligns with theoretical expectations. We agree that this demonstration primarily serves as a "sanity check" for the method’s design. However, as noted in Section 3.4 of the paper, it also provides a lower bound on the usable batch size for stable alignment. We could expand on this point in the revised paper to emphasize the practical implications of this analysis.
>
> - **Cosine Similarity Tracking as an Ablation**: To enhance the interpretive value of this experiment, we plan to follow the reviewer’s suggestion and extend it by tracking the cosine similarity between the projected error and the true gradient throughout training on real datasets and comparing this alignment with DFA.
>
> ## 4. Justification for Activity Perturbation vs. Weight Perturbation
>
> The choice of activity perturbation over weight perturbation is a deliberate design choice in GrAPE, driven by theoretical and practical considerations:
>
> - **Reduced Variance and Dimensionality**: Perturbing the activity space (rather than weight space) operates in a lower-dimensional space, resulting in reduced variance in gradient estimation, which is crucial for stable alignment in high-dimensional networks. Weight perturbation introduces noise over a much larger parameter space, potentially destabilizing the gradient approximation. Basically, for a given weight matrix $ W $, representing the connections between a layer of size n and a layer of size m ; the number of parameters is $ O(n \times m) $, while the number of activations is $ O(n) $, where $ n $ is the number of neurons in the layer. This makes the activity space a more stable and efficient choice for perturbation.
>
> - **Alignment with the weight gradient**: The reviewer rightfully explains that aligning the feedback matrix with the weight gradient seems more intuitive. However, and we will clarify this in the revised version of the paper, this would mean to align $B_l e$ with the gradient. In the case of a null error, the alignment would also be null, which would prevent the feedback matrix from being updated. The goal of GrAPE is to align the matrix *conveying* the error signal e with the jacobian, which is the true gradient when multiplied by the error signal e.

---

> ### Author Response · Authors · 2024-11-18
> **Answer to weakness 5**
>
> ## 5. Runtime Comparison with Other Methods
>
> We appreciate the reviewer’s suggestion to include runtime comparisons. Calculating forward gradients via Jacobian-vector products (JVPs) does introduce an additional overhead compared to DFA. Based on preliminary tests, GrAPE incurs an approximate 15-20\% increase in computation time per epoch compared to DFA, though this may vary with network depth and batch size. It has to be noted that the JVP calculations used in GrAPE are based on Pytorch's Forward-Mode AutoDifferentiation which is still in beta.
>
> ---
>
> We hope these clarifications address the reviewer’s concerns and we are working on adding the suggested changes and additional results in the revised version of the paper shortly.
>  We are open to further feedback and discussion with the reviewer to enhance the quality and presentation of our work.

---

> ### Comment · Reviewer_R1hf · 2024-11-22
> **Paper motivation and answer to Weakness 1**
>
> The reviewer sincerely thank the authors for their detailed response and would like some further clarifications on some points.
>
>
> # Clarification on Eq.(6) and Mini-Batch behavior
> If the reviewer understands correctly, in an ideal setting where authors would perform several forward gradient passes and using eq.(6) to update the matrix $B_l$ each time, it should converge to
> $$
> B_l^* = \mathbb{E}[\nabla \mathcal{L}_l(x)]
> $$
> and thus favor an alignment with respect to the expected Jacobian.
> Can the author confirm that this intuition is correct ?
> If true, it would be a good qualitative information to explicitly give in the main body, with a formal proof deferred in the appendix if authors can derive it.

---

> > ### Author Response · Authors · 2024-11-23
> > **Convergence of feedback matrices with multiple forward passes**
> >
> > We would like to thank the reviewer. The given insights are correct and adding them in the paper will give a direct intuition of our method which would further strengthen it.
> > This will be made clear in the main paper.
> > In a ideal setting, it should converge to a "rescaled" gradient as the feedback matrix has a constant norm.

---

> ### Comment · Reviewer_R1hf · 2024-11-22
> **Answer to Weaknesses 2-4**
>
> # 2. Ablation Studies and Compatibility with Batch-Normalization
>
> - **feedback matrix rescaling:** an ablation on the rescaling factor would indeed be much appreciated as the authors introduces a novel learning dynamics which needs to be probed empirically. This can be deferred in the appendix.
> - **Batch-Normalization Compatibility:** this ablation is much more crucial and should be mentioned in the main body, or at least a solid appendix detailing experiments on this particular matter. Batch-normalization introduces several issues, such as a dependence of computations among samples which hampers parallelization potential and requires serious implementation engineering in certain scenarios. As the authors says at line 490 that they notice a positive effect, this needs to be quantified with empirical evaluations.
>
> # 3. Analysis of Cosine Similarity over Time
> The reviewer appreciate this effort, and would like to know if the authors could provide some cosine similarity figures before the end of the rebuttal. If not, it is understandable, but the reviewer is very curious. The Zoutendijk theorem seems to claim that a positive alignment is enough, but from experience, the reviewer knows that below $10^{-3}$, it becomes seriously challenging to obtain competitive optimization performances. Having $10^{-2}$ might be low but acceptable to the reviewer's opinion
>
> # 4. Justification for Activity Perturbation vs. Weight Perturbation
>
> - **Reduced Variance and Dimensionality:** the reviewer is pretty sure that the reduced dimensionality argument does not apply to the early layers of VGG-16. Indeed, convolutional architectures often have fewer parameters than input or output dimensions, which makes weight perturbation more stable in theory. The theory actually predicts that one can choose to perform activity or weight perturbation depending on the dimensionality of the output and parameter space for a given layer. The authors are free to confirm or give a counter argument.
>
> - **Alignment with the weight gradient:** the reviewer is a bit puzzled by this argument... If the error is null, why would we even need to further train the model ? It is strange to take this as a motivating example for activity perturbation over weight perturbation.

---

> > ### Author Response · Authors · 2024-11-23
> > **Further Answers**
> >
> > ## 2. Ablation Studies
> > ### Feedback matrices rescaling
> > We will provide ablation studies regarding this point in the Appendix.
> > Preliminary results show a detrimental effect of removing this rescaling, reaching the following results when training AlexNet on CIFAR 100. It is also notable that depending on the chosen learning rate and scheduler, the performance might also change.
> >
> > | DFA  | Grape with rescaling |Grape w\\o rescaling |
> > |----------------|-----------------------|-----------------------|
> > |29.75\%|40.43\%|32.31\%|
> >
> > ### Batch Normalization
> > You are right to point out that batch normalization introduced a dependence of computations among samples. This is especially true in standard settings where the scale and shift parameters of the batch normalization are trained. In the presented results we set affine to False so that they are fixed and thus non trainable, removing this need for sequentiality.
> >
> > The following results come from previous runs training AlexNet on CIFAR 100 with and without batch normalization, further experiments might indicate change in results.
> > | DFA  | Grape with BN|Grape w\\o BN |
> > |----------------|-----------------------|-----------------------|
> > |29.75\%|40.43\%|34.7\%|
> >
> > We will provide a complete set of results in the Appendix in the final version of the paper.
> >
> > ## 3. Cosine Similarity Figures
> > We provided a toy example with fixed gradient in the Appendix (Figure 4) to highlight the dynamics of our proposed update on the cosine similarities. We are working on the same kind of figure but this time with the proposed experiments settings.
> >
> > ## 4.Choice of perturbation
> > ### Reduced variance and dimensionality
> > The reviewer indeed has a very good point that we had overlooked when it comes to convolutional layers. The shared weights structure would allow to get a less variant gradient approximation when perturbing the weights rather than the activations in this case.
> > Although we did use standard techniques to reduce variance (moving average, averaging over batches), finding a less variant estimation depending on the layers inner structure is very astute.
> >
> > However, it might be difficult to incorporate further experiments to empirically verify if performing a chosen perturbation setting depending on the dimensionality of the output and parameter space for every layer would help reach better minima in the remaining time we have. This might however be an interesting idea for future work.
> >
> > ### Alignment with the weight gradient
> > It is true our argument is not clear enough. Furthermore, we actually gave that example to explain why we align $B_l$ with the jacobian and not $B_l e$ with the gradient.
> > - Let us clarify the example we gave:
> > We were considering the setup where the network has perfectly memorized a given sample.
> > With perfect prediction on a given sample, the error would be equal to 0, which would not mean however that the overall training error is null (so the training has to continue).
> > \\In that case, the gradient is also null but the jacobian is not. The weights of the network will not be updated because it perfectly predicted the output, however the feedback matrices should align with the jacobian.
> > - The choice of activity perturbed gradient over weight perturbed gradient was actually made because in the general case of dense layers, the variance is lower but as mentioned by the reviewer, weight perturbed gradient might actually be more accurate in the case of convolutional layers. A hybrid solution would be ideal but might be out of the scope of the paper.

---

> > > ### Author Response · Authors · 2024-11-25
> > > **Theoretical Analysis and additional experiments**
> > >
> > > When advancing the theoretical proof, we found we needed a slight modification of the update rule to ensure convergence.
> > > This modification was adapted from the idea of reviewer xxPy.
> > >
> > > Let us remind the notations:
> > > - $ B $ is the feedback matrix.
> > > - $ \eta $ is the learning rate.
> > > - $ \cos(w) $ is the cosine similarity between $ B $ and $ g_l $, computed as:
> > >   $
> > >   \cos(w) = \frac{B \cdot g_l}{\|B\| \|g_l\|}.
> > >   $
> > > - $ \tilde{g} $ is the gradient $ g_l $, rescaled to match the norm of $ B $:
> > >   $
> > >   \tilde{g} = g_l \cdot \frac{\|B\|}{\|g_l\|}.
> > >   $
> > > to ensure the added component has a constant impact on the direction update of the feedback matrix.
> > >
> > >
> > > We consider two alternatives:
> > > - $B \leftarrow B - \eta (1 - \cos(w)) B + \eta \cos(w) \tilde{g},$ denoted as Alternative 1
> > > - $B \leftarrow (1 - \eta ) B + \eta  \tilde{g},$ denoted as Alternative 2
> > >
> > > We illustrate Figure 5 (in the Appendix), the original update when compared to the two proposed alternatives.
> > > The setup of Figure 5 is the following: $g_l$ is fixed and we compare the impact of the different learning rule on the cosine similarity of $B$ with $g_l$. The matrices dimensions are (200x200). The mean across 10 random initialisations of $B$ and $g_l$ is represented.
> > >
> > > It shows that **the original rule only performs marginally worse than the alternative 1**, but follows the same trend, while alternative 2 is definitely worse than the two others, with final cosine similarity (after 50 training steps) of 0.
> > > **We will add experiments using alternative 1 to the final version of the paper.**
> > >
> > > ---
> > > Alternative 1 further allows us to derive theoretical results in an ideal and easy setting:
> > >
> > > Let us rewrite
> > > $B = B_{\parallel} + B_{\perp},$
> > > where:
> > > - $ B_{\parallel} = \frac{(B \cdot g_l)}{\|g_l\|^2} g_l $ is the parallel component of $B$ to $g_l$
> > > - $ B_{\perp} = B - B_{\parallel} $ is the orthogonal component of $B$ to $g_l$
> > >
> > >
> > > The update rule for the parallel component is:
> > > $
> > > B_{\parallel}^{t+1} = B_{\parallel}^t - \eta (1 - \cos(w)) B_{\parallel}^t + \eta \cos(w) \tilde{g}.
> > > $
> > > Which can read:
> > > $
> > > B_{\parallel}^{t+1} = (1 - \eta (1 - \cos(w))) B_{\parallel}^t + \eta \cos(w)  \tilde{g}.
> > > $
> > >
> > > The update rule for the orthogonal component is:
> > > $
> > > B_{\perp}^{t+1} = B_{\perp}^t - \eta (1 - \cos(w)) B_{\perp}^t.
> > > $
> > > Since $ \cos(w) \leq 1 $, the factor $1 - \eta (1 - \cos(w)) $ reduces $ B_{\perp} $ with each step, causing it to decay exponentially:
> > > $$
> > > B_{\perp}^{t+1} = (1 - \eta (1 - \cos(w))) B_{\perp}^t.
> > > $$
> > >
> > > We can write:
> > > $
> > > B^{t+1} = B_{\parallel}^{t+1} + B_{\perp}^{t+1}.
> > > $
> > > As $ t \to \infty $:
> > > - $ B_{\perp} \to 0 $, because the orthogonal component decays exponentially.
> > > - $ B_{\parallel} $ aligns with $\tilde{g}$, since:
> > >   $$
> > >   B_{\parallel}^{t+1} \to \frac{B \cdot g_l}{\|g_l\|^2} g_l.
> > >   $$
> > >
> > > Thus, $B \to \tilde{g}$, achieving full alignment.
> > >
> > > ---
> > > As a conclusion, alternative 1 inspired by the reviewer xxPy remarks ensures:
> > > 1. Exponential decay of orthogonal components.
> > > 2. Preservation and alignment of parallel components.
> > > 3. Smooth and stable convergence to $\tilde{g}$, meaning a total alignment with $g_l$.
> > >
> > > However as illustrated Figure 5, it only performs marginally better than the original learning rule that does not offers such guarantees. We will test and compare on the paper's experimental setup if adding a gradient component enhances the original rule's performance.

---

> > > > ### Comment · Reviewer_R1hf · 2024-12-02
> > > > **Final review**
> > > >
> > > > I appreciate the authors effort into clarifying their submitted work during this rebuttal period. I tried to follow as closely as possible the discussion with reviewer xxPy to clarify many aspects that I did not understand. For example, I was sure that the cosine was a scalar and not a vector, and thus I did not expect an Hadamard product in eq.(6). While the reviewer is enthusiast about the reported numerical results, much improvement is needed with respect to the clarity of the paper. The reviewer did his best to see through any typo, but this require substantial amount of time and I am still not able to tell whether the mathematical details are correct for reproducibility. Therefore, I am keeping my score as it is.

---

### Author Response · Authors · 2024-11-18
**General Comment**

Dear Reviewers,

Thank you for your insightful comments and interesting questions regarding our work.

We have provided dedicated and detailed responses addressing each of the issues you pointed out.

We are open to and welcome further discussion to help us improve our paper and to answer questions.

---

### Meta-Review · Area_Chair_1ZXH · 2024-12-14

**Metareview:**

Reviewers acknowledged the authors’ efforts to improve backpropagation-free learning and they appreciated the theoretical basis and experiments. However, they expressed confusion and concern regarding the clarity and correctness of the update rule and its notation, the dimensional consistency of the involved matrices, and the exact meaning of “alignment.” Multiple requests for clarification revealed issues in the paper’s presentation and theoretical justification. While reviewers found the idea promising and potentially impactful, they concluded that the current manuscript requires significant revision for clarity, rigorous notation, and additional ablation studies before it can be fully endorsed.

**Additional Comments On Reviewer Discussion:**

During the rebuttal phase, extensive discussion revealed notational inconsistencies that the reviewers were correct to note. Based on the continued difficulty of two reviewers in understanding the algorithm in the manuscript, it is advised to significantly revise the writing and resubmit.

---

### Decision · Program_Chairs · 2025-01-22

Reject